# Cellular and genetic drivers of RNA editing variation in the human brain

Winston H. Dredge[1,2,3,4,7], Junhao Li [5,7], Xuanjia Fan[1,2,3,4], Alexey Kozenkov[1,6], Matthew Lalli[1,3], Shahrukh Khalique[1,6], Stella Dracheva[1,6], Eran A. Mukamel [5] & Michael S. Breen [1,2,3,4✉]

Posttranscriptional adenosine-to-inosine modifications amplify the functionality of RNA molecules in the brain, yet the cellular and genetic regulation of RNA editing is poorly described. We quantify base-specific RNA editing across three major cell populations from the human prefrontal cortex: glutamatergic neurons, medial ganglionic eminence-derived GABAergic neurons, and oligodendrocytes. We identify more selective editing and hyper-editing in neurons relative to oligodendrocytes. RNA editing patterns are highly cell type-specific, with 189,229 cell type-associated sites. The cellular specificity for thousands of sites is confirmed by single nucleus RNA-sequencing. Importantly, cell type-associated sites are enriched in GTEx RNA-sequencing data, edited ~twentyfold higher than all other sites, and variation in RNA editing is largely explained by neuronal proportions in bulk brain tissue. Finally, we uncover 661,791 cis-editing quantitative trait loci across thirteen brain regions, including hundreds with cell type-associated features. These data reveal an expansive repertoire of highly regulated RNA editing sites across human brain cell types and provide a resolved atlas linking cell types to editing variation and genetic regulatory effects.

[1] Department of Psychiatry at Mount Sinai, New York, NY 10029, USA. [2] Department of Genetics and Genomic Sciences at Mount Sinai, New York, NY 10029, USA. [3] Seaver Autism Center for Research and Treatment at Mount Sinai, New York, NY 10029, USA. [4] Pamela Sklar Division of Psychiatric Genomics at Mount Sinai, New York, NY 10029, USA. [5] Department of Cognitive Science, University of California, San Diego, La Jolla, CA 92037, USA. [6] James J Peters VA Medical Center, Bronx, NY 10468, USA. [7] These authors contributed equally: Winston H. Dredge, Junhao Li. ✉email: michael.breen@mssm.edu

The complexity of the central nervous system (CNS) is largely coordinated through multiple layers of transcriptional regulation, generating functionally distinct RNA molecules with specialized posttranscriptional modifications[1,2]. Adenosine to inosine (A-to-I) editing is abundant in the human brain and predicted to occur at millions of locations across the genome[3,4]. A-to-I editing occurs at single isolated adenosines (selective editing) as well as in extended regions with multiple neighboring adenosines (RNA hyper-editing)[5–7] and is catalyzed by adenosine deaminase acting on RNA (ADAR) enzymes. These base-specific changes exponentially amplify RNA sequence diversity and expand the functionality of many brain-expressed genes, by allowing the same coding sequence to produce different mRNA and products. Although the growth of RNA-sequencing datasets has increased the catalog of known editing sites, the functional relevance of most sites remains unknown. We reasoned that sites with precisely regulated differences in RNA editing levels across brain cell types or brain regions signal a potentially critical role in supporting the functional diversity of brain circuits. Moreover, dissecting the genetic regulation of RNA editing at these sites gives insight into their potential role in healthy and diseased brains.

In the CNS, RNA editing regulates neuronal transcription, splicing, and subcellular localization of mRNA transcripts[1,8–11]. RNA editing in protein-coding regions can result in recoding specific amino acids, which influences essential neurodevelopmental processes, including actin cytoskeletal remodeling at excitatory synapses[1,12], regulation of gating kinetics of inhibitory receptors[1,13], and modulation of neurotransmission at inhibitory synapses[1,14]. RNA editing sites are dynamically regulated throughout human cortical development[15,16], with marked increases in editing levels occurring between mid-fetal development and infancy. These profiles are conserved in nonhuman primates and murine models of cortical development, indicating an evolutionarily selected function[12,15]. Moreover, widespread changes in RNA editing are linked to several neuropsychiatric and neurodevelopmental disorders[17–22]. Yet, the cellular specificity of RNA editing sites in the human brain remains largely unknown due to the lack of cell type-specific studies.

The vast majority of RNA editing sites have been detected in bulk brain tissue, a mixture of dozens of neuronal and glial cell types with distinct transcriptional, and potentially epitranscriptional, programs. In the mouse brain, editing levels are higher in neurons relative to glial cells, and these trends are consistent across brain development[23,24]. In *Drosophila*, RNA editing has been explored across several neuronal populations, each containing a unique editing signature composed of distinct site-specific editing levels in neuronal transcripts[25]. In the human brain, the challenge of purifying specific cell populations has so far prevented broad-based analysis of cell type-specific RNA editing. Single-cell RNA-sequencing has been applied in a modest number of cells[26] ($n^{cells} = 268$), indicating that highly edited sites in individual cells often go undetected by bulk brain RNA-seq. The systematic identification of bona fide, high-confidence cell type-associated RNA editing sites in the human brain is critical for understanding the scope and specificity of this layer of cell regulation.

In addition to cell-specific factors, common genetic variation has also recently emerged as an important regulator of RNA editing levels in the brain[17,27]. The integration of paired genomic and transcriptomic data can identify RNA editing quantitative trait loci (edQTLs). It is estimated that anywhere between ~10–30% of RNA editing sites identified in bulk tissue samples of the human brain are regulated by common genetic variants[17,27,28], and this number will continue to grow with increasing sample sizes. However, there has been no investigation of cell type-associated edQTLs in the human brain.

Here we sought to expose the main cellular and genetic drivers of RNA editing variation in the human brain. We first quantified RNA editing among three major cell populations purified from the adult prefrontal cortex (PFC)—the brain region critical for cognition, memory, and executive function and is broadly implicated in neuropsychiatric illness[29]. The PFC contains two major neuronal populations, excitatory glutamatergic (GLU) and the inhibitory GABAergic interneurons, which account for about 80% and 20% of all cortical neurons, respectively[30]. Medial ganglionic eminence (MGE)–derived interneurons comprise ~60–70% of all cortical GABAergic neurons and contain parvalbumin- and somatostatin-expressing interneurons, which have been implicated in neurodevelopmental and neuropsychiatric disorders[31,32]. Oligodendrocytes (OLIG) are the major glial cell type in the central nervous system that provides support and myelin-based insulation to axons[33]. Here, GLU, MGE-GABA, and OLIG populations were isolated from nine donors using fluorescence-activated nuclei sorting (FANS), followed by transcriptomic analysis by RNA-seq[34]. By combining our high-resolution cell type-specific data with complementary single nucleus RNA-seq, as well as bulk RNA-seq from multiple brain regions from the GTEx project, we provide comprehensive analysis and validation of cell type-specific RNA editing across the human brain, including key genetic regulators of editing quantitative trait loci (edQTLs). Overall, our study suggests that RNA editing plays a critical role in supporting the diverse molecular identities of brain cell types.

## Results

**Global editing rates in MGE-GABAergic interneurons, glutamatergic neurons, and oligodendrocytes.** To better understand the differences in RNA editing between MGE-GABA, GLU, and OLIG cells, we computed an *Alu* editing index (AEI) as a global measure of site-selective RNA editing activity for each cell type (*see Methods*). The AEI is defined as the ratio of the total number of A-to-G edited reads over the total coverage of all adenosines in *Alu* elements across the transcriptome. The AEI was higher in neurons compared to OLIG (Cohen's d = 2.58, $p = 1.2 \times 10^{-6}$, linear regression) and elevated in MGE-GABA relative to GLU (Cohen's d = 1.46, $p = 0.009$, linear regression) (Fig. 1a and Supplementary Data 1). Given that the vast majority of RNA editing occurs in *Alu* elements and nearly all adenosines in *Alu* repeats are targeted by ADARs, we queried ADAR expression levels and identified higher expression of *ADAR1* in MGE-GABA and GLU cells relative to OLIG (Cohen's d = 4.34, $p = 8.2 \times 10^{-11}$, linear regression), higher expression of *ADAR2* in MGE-GABA neurons relative to GLU and OLIG (Cohen's d = 1.96, $p = 0.0002$, linear regression), and higher *ADAR3* expression in OLIG relative to neurons (Cohen's d = 1.78, $p = 0.001$, linear regression) (Fig. 1b). Notably, variation in the AEI was positively associated with *ADAR2* ($R^2 = 0.68$) and *ADAR1* ($R^2 = 0.64$) expression and negatively associated with *ADAR3* ($R^2 = 0.33$) (Fig. 1c).

We also computed a global metric of A-to-G hyper-editing, defined as consecutive editing across many neighboring adenosines within an extended region in the same transcript, which leverages unmapped RNA reads (*see Methods*) (Supplementary Data 1). There were ~4 times more RNA hyper-editing sites in MGE-GABA (μ = 266,621 sites) and GLU (μ = 251,790 sites) neurons than in OLIG (μ = 65,716 sites) (Cohen's d = 1.83, $p = 0.0002$, linear regression) (Fig. 1d). To minimize technical variability and facilitate a direct comparison across all cell types, we normalized the hyper-editing signal to the number of mapped

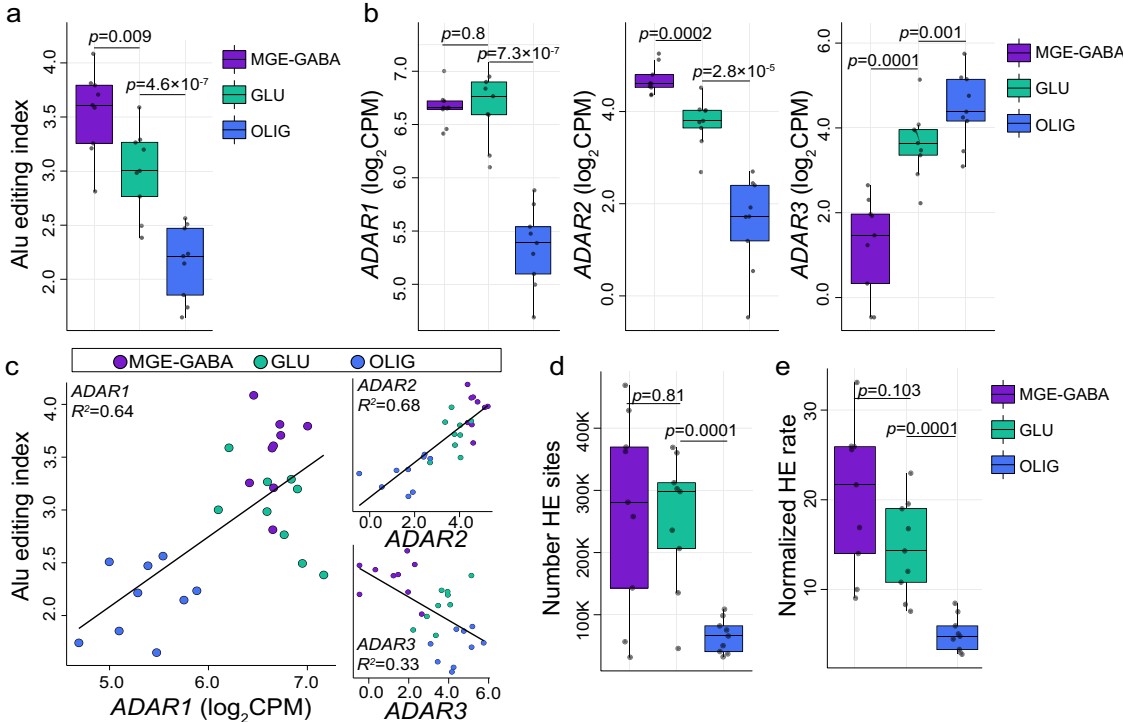

**Fig. 1 Global selective editing and hyper-editing in purified cortical cell types. a** The *Alu* editing index (AEI) (y-axis) and **b** normalized expression for *ADAR1*, *ADAR2*, and *ADAR3* (y-axis) measured for each FANS-derived cell population from the human adult prefrontal cortex (PFC, $n = 9$ biologically independent samples): MGE-GABA, GLU, and OLIG populations. **c** The amount of AEI variance explained ($R^2$, y-axis) by *ADAR* expression. **d** The total number of hyper-editing (HE) sites (y-axis) and **e** the normalized HE rate (y-axis) across all cell types. All box plots show the medians (horizontal lines), upper and lower quartiles (inner box edges), and 1.5 × the interquartile range (whiskers). The normalized HE rate is defined as the number of HE sites detected per million mapped (MM) bases, computed by dividing the total number of HE sites over the total number of MM bases per sample and multiplying the result by one million. Two-sided linear regression computed significance in mean differences between MGE-GABA vs. GLU and GLU vs. OLIG. No adjustments were made for multiple comparisons. In all panels, MGE-GABA vs. OLIG were deemed statistically significant ($p < 0.05$). Cohen's *d* was used as a measure of effect size pertaining to differences in the AEI and HE signal between cell types.

bases per sample and again observed a preponderance of hyper-editing in neurons relative to OLIG (Cohen's d = 2.09, $p = 4.4 \times 10^{-5}$, linear regression) (Fig. 1e). Normalized hyper-editing rates were also positively associated with *ADAR1* ($R^2 = 0.33$) and *ADAR2* ($R^2 = 0.54$), but negatively associated with *ADAR3* ($R^2 = 0.21$) (Supplementary Fig. 1). Hyper-editing sites commonly occurred in introns and 3′UTRs (Supplementary Data 1) and demonstrated enrichment for a local RNA editing sequence motif whereby guanosine is depleted −1 bp upstream and enriched +1 bp downstream the target adenosine (Supplementary Fig. 1D, E); consistent with the previous reports[7,15], thus further validating the accuracy of the hyper-editing approach. Taken together, we show that global selective editing and RNA hyper-editing are more prevalent in MGE-GABAergic than glutamatergic neurons, followed by OLIG cells, and this variation is mainly driven by ADAR expression.

**Identification and annotation of bona fide site-selective RNA editing sites**. To catalog high-confidence bona fide selective editing sites, we combined de novo calling with a supervised approach (*see Methods*). All sites were subjected to rigorous downstream filtering and quality control. In brief, thresholds were set to control total read coverage (>10 supporting reads), minimum edited read coverage (>3 supporting edited reads), and a minimum editing ratio (at least 5%). We further required that sites meet these criteria in at least eight out of nine donors. Sites in homopolymeric and blacklisted regions of the genome were discarded, along with sites marked as common genomic variants (Fig. 2a). Overall, 189,229 cell type-associated RNA editing

sites were identified, including a total of 107,998 sites on 4781 genes in MGE-GABA, 109,734 sites on 4935 genes in GLU, and 64,374 sites on 3469 genes in OLIG cell populations (Supplementary Data 2), discovery rates consistent with higher global editing levels in neurons (Fig. 1). Approximately 36% of MGE-GABA, ~35% of GLU, and ~55% of OLIG sites classify as cell type-specific as they were uniquely detected in one cell type, and their detection rates were largely explained by cell type-specific gene expression and read coverage differences (Supplementary Fig. 2).

To validate the accuracy of our approach, we annotated all events and observed consistent hallmarks of RNA editing. First, the vast majority of sites were A-to-G edits (~86%) (Fig. 2b) and resided within *Alu* elements (~68%) (Fig. 2c). Second, ~73% of editing events were detected in introns, while only a small fraction impacted protein-coding regions (~0.67%) (Fig. 2d). Third, while most RNA editing sites were known events cataloged in the REDIportal database (Fig. 2e), we also identified thousands of novel (not in catalog) A-to-G events, including 20,929 sites in MGE-GABA, 18,452 sites in GLU and 7366 sites in OLIG. Fourth, we confirmed a common local sequence motif for all known and not in catalog A-to-G editing events, whereby guanosine is depleted −1 bp upstream and enriched +1 bp downstream the targeted adenosine, as previously reported (Fig. 2f, g). Notably, not in catalog A-to-G sites displayed significantly more supporting read coverage compared to known sites (Fig. 2h) and exhibited consistent editing rates of ~37% on average across all cells (~3% less than known sites, $p < 2.16 \times 10^{-20}$, linear regression) (Fig. 2i). Further, not in

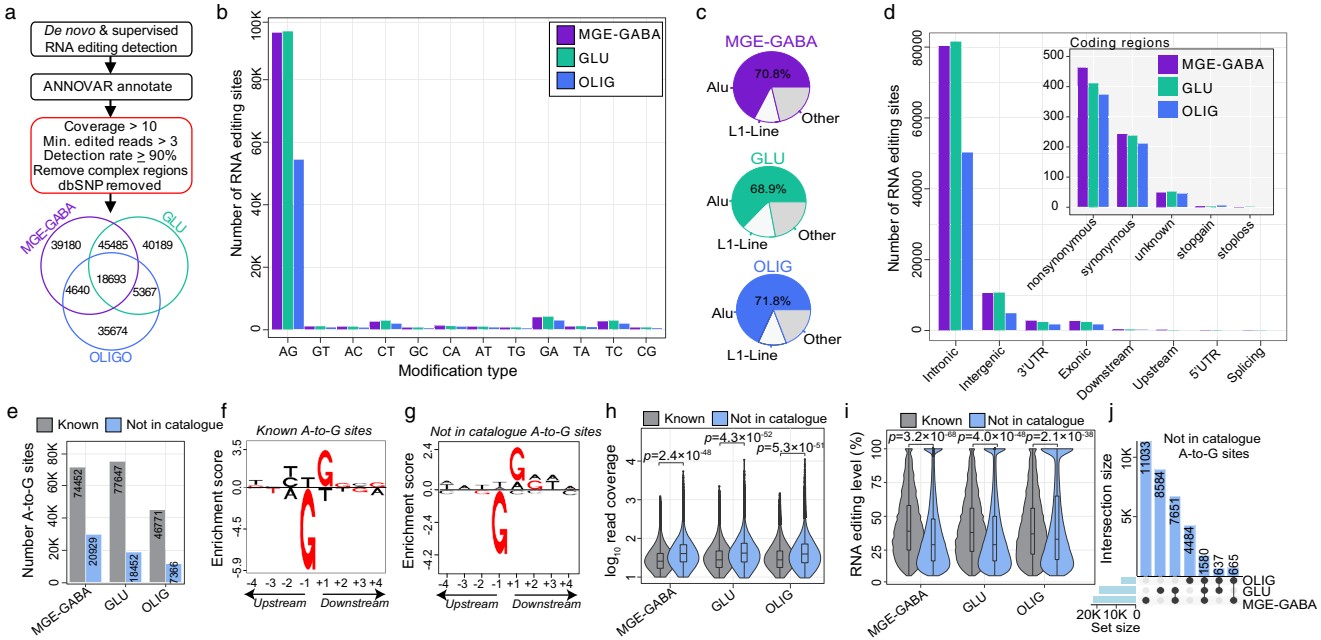

**Fig. 2 Identification and annotation of cell-specific RNA editing sites. a** Workflow for detecting high-confidence cell type-associated editing sites combining de novo calling and supervised RNA editing detection. The total number of sites detected per cell type are displayed in the Venn diagram. Data in all panels were derived from the human PFC (*n* = 9 biologically independent samples). **b** The total number of RNA editing sites by modification type. **c** The fraction of sites that map to *Alu* and L1-line elements relative to all other elements. **d** The total number of editing sites by genic region. The inset plot shows the breakdown in coding regions. **e** The number of A-to-G editing sites (y-axis) were grouped based on catalog annotation (known or not in catalog) and independently numerated per cell type. Local sequence motif enrichment analysis for both **f** known and **g** not in catalog sites depicting a depletion of guanosine -1bp upstream and enrichment +1 bp downstream of the target adenosine. Known and not in catalog sites for each cell population according to **h** read coverage (log₁₀, y-axis) and **i** editing level (%, y-axis). Two-sided Mann–Whitney *U*-tests computed significance in mean differences in read coverage differences and editing rates between known and not in catalog sites within MGE-GABA, GLU, and OLIG populations and associations with a *p* value <0.05 were deemed significant. Box plots show the medians (horizontal lines), upper and lower quartiles (inner box edges), and 1.5 × the interquartile range (whiskers). **j** An upset plot highlights the cell type convergence/divergence of not in catalog A-to-G sites.

catalog sites were commonly detected on transcripts expressed at low-to-moderate levels (Supplementary Fig. 3). Moreover, 7326 not in catalog A-to-G sites were validated across two or more cell types (Fig. 2j). We also identified more than 20,000 not in catalog sites with substitution types other than A-to-G. Of these, ~42% were C-to-T and G-to-A edits, which we treated as provisional (Supplementary Data 2).

**Partitioning the variance in RNA editing levels explained by known factors**. We studied 15,221 A-to-G sites detected across all three cell types and all donors to quantify the fraction of RNA editing variance explained by eight known biological and technical factors. Collectively, these factors explained ~28% of RNA editing variation. Differences between cell types had the largest genome-wide effect, explaining a median of ~8.3% of the observed variation, followed by differences in chronological age (~6.7%), pH (~3.5%), *ADAR2* (~1.8%), and *ADAR1* expression (~1.5%) (Supplementary Fig. 4A). Donors as a repeated measure had a small but detectable effect for ~6% of editing sites. As expected, principal component analysis accurately distinguished MGE-GABA and GLU neurons from OLIG along the first PC1, explaining 29.2% of the variance (Supplementary Fig. 4B). Using a linear regression model, we also cataloged a total of 6765, 7703, and 2540 selective editing sites that were significantly associated with *ADAR1*, *ADAR2*, or *ADAR3* expression, respectively, after adjusting for repeated measures (FDR < 0.05), respectively (Supplementary Fig. 4C and Supplementary Data 3).

In addition to ADARs, several RNA-binding proteins (RBPs) can also act as global mediators of RNA editing. We found a total of 470 RBPs were differentially expressed (FDR < 0.05, log fold-

change > 0.5), and roughly half (~49%) were more highly expressed in OLIG relative to GLU and MGE-GABA (Supplementary Fig. 5A, B). A total of 170 OLIG-specific RBPs, including NOP14, PTEN, and DYNC1H1 were negatively correlated with global editing activity, while 161 neuron-specific RBPs, including MOV10, CELF4, and FMRP had a positive association with global editing rates (Supplementary Fig. 5C and Supplementary Data 3). We further examined whether FMRP binding sites were enriched near MGE-GABA, GLU, and OLIG editing sites using existing data for enhanced ultraviolet crosslinking and immunoprecipitation (eCLIP) across two technical replicates in the human frontal cortex[18]. FMRP eCLIP peaks were significantly enriched for MGE-GABA (Rep₁, *Z*-score = 9.3; Rep₂, *Z*-score = 4.7), GLU (Rep₁, *Z*-score = 6.2; Rep₂, *Z*-score = 3.7), and OLIG sites (Rep₁, *Z*-score = 7.4; Rep₂, *Z*-score = 2.8) (Supplementary Fig. 5D, E). Thus, these RBPs may work alongside ADARs as *trans* regulators of editing levels in brain cell populations.

**Cell type-enriched RNA editing sites**. To identify quantitative differences in RNA editing levels among sites detected across two or more cell types, we computed three pairwise comparisons (i.e., MGE-GABA vs. GLU, MGE-GABA vs. OLIG, GLU vs. OLIG) and adjusted each linear model for the possible influence of postmortem interval (PMI), age, and donor as a repeated measure (Fig. 3a and Supplementary Data 4). Overall, 13,104 cell type-enriched sites displayed significantly higher editing in one cell population relative to another. The majority of these sites were more highly edited in MGE-GABA and GLU relative to OLIG (Fig. 3b), are previously reported editing sites, and commonly mapped to introns and 3′UTRs. (Fig. 3c). Notably, after adjusting

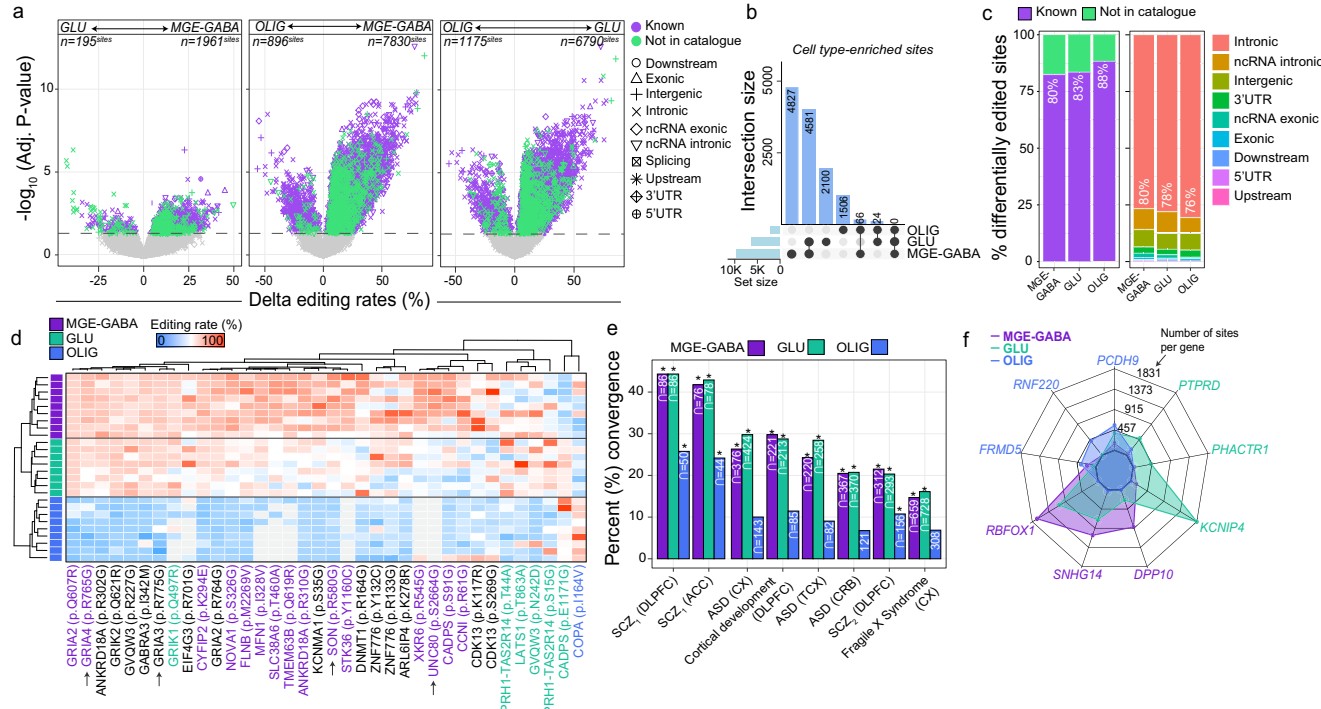

**Fig. 3 Cell type differential editing, recoding sites and gene-level enrichment. a** Differential RNA editing compared GLU vs. MGE-GABA (left), OLIG vs. MGE-GABA (center), and OLIG vs. GLU (right) ($n = 9$ biologically independent samples per cell type). Change in editing levels (%, x-axis) compared to the strength of significance ($-\log_{10}$ Adj. p value). The horizontal axis marks an adjusted p value of 0.05. RNA editing sites are uniquely shaped by the genic region and colored by catalog annotation. **b** UpSet plot reporting counts for all cell type-enriched sites are reported with their overlaps. **c** The percentage (%, y-axis) of differentially edited sites by catalog annotation (left) and genic region (right). **d** Heatmap displaying differential editing levels at recoding sites across MGE-GABA (blue), GLU (purple), and OLIG (green) cells. The gray color indicates sites that were not detected in a given sample. Arrows (x-axis) indicate sites validated by independent methods (Supplementary Fig. 9). **e** The percentage of previously reported altered editing sites in bulk brain tissues with cell type-associated features (%, y-axis). Dysregulated RNA editing sites in bulk brain tissue were collected from the dorsolateral prefrontal cortex (DLPFC), anterior cingulate cortex (ACC), cortex (CX), temporal cortex (TCX), cerebellum (CRB), schizophrenia (SCZ; $SCZ_1$ CommonMind Consortium; $SCZ_2$ NIMH HBCC cohort), autism spectrum disorder (ASD), cortical development (Hwang et al., 2016). One-sided Fisher's exact test and an estimated odds ratio was used to compute the significance of each overlap and a p value <0.05 (*) was deemed significant. Exact p values for all tests are reported in Supplementary Data 4. **f** Spider plot of the top three genes for each cell type that displays enrichment for RNA editing sites after accounting for gene length.

for *ADAR1* and *ADAR2* expression, we found substantially fewer differentially edited sites ($n = 192$ sites), suggesting the differential editing at the majority of these sites depends on ADAR activity (Supplementary Data 4). Cell type-enrichment differences in RNA editing rates explained ~7, ~7, and ~2% cell-specific differential gene expression in MGE-GABA ($n^{genes} = 775$), GLU ($n^{genes} = 660$), and OLIG ($n^{genes} = 378$) populations, respectively (Supplementary Fig. 7). Moreover, a small subset of cell type-enriched RNA editing sites were predicted to be splice altering in MGE-GABA ($n^{sites} = 28$), GLU ($n^{sites} = 23$), and OLIG ($n^{sites} = 23$) populations (Supplementary Data 4).

Functional annotation revealed that cell type-enriched sites in each cell type, predominantly those in introns, were associated with genes involved in neuronal differentiation and cell adhesion, but were also enriched for unique processes (Supplementary Data 4). MGE-GABA sites were uniquely enriched for genes implicated in trans-synaptic signaling, MAPK signaling, as well as genes at the postsynaptic density (Supplementary Fig. 6). GLU sites were uniquely enriched for genes implicated in chromatin organization and interferon signaling. OLIG sites were uniquely enriched for genes associated with *N*-acetyltransferase activity and methylated histone binding.

A small fraction of differentially edited sites was cataloged as RNA recoding events (Fig. 3d, ~0.4%, $n = 36$ sites), which alter amino acid states and displayed increased conservation relative to

sites in other genic regions (Supplementary Fig. 8 and Supplementary Data 5). These recoding events were mainly more highly edited in neurons relative to OLIG and included several well-known sites involved in the tight regulation of $Ca^{2+}$ permeability (Q→R in *GRIA2*) actin cytoskeletal remodeling at excitatory synapses (K→E in *CYFIP2*) and gating kinetics of inhibitory receptors (I→M in *GABRA3*) (Fig. 3d). Twelve recoding sites were more frequently edited in MGE-GABA compared to GLU, including an R→G site in cyclin-I (*CCNI*), a K→R site in mitofusin 1 (*MFN1*), a E→G site in calcium-dependent secretion activator (*CADPS*). Notably, an I→V site in coatomer subunit alpha (*COPA*) was more highly edited in OLIG relative to neurons. We tested the cellular specificity for four recoding sites using an independent method (site-specific PCR amplification of regions harboring editing sites followed by sequencing) applied to orbitofrontal cortex samples from six independent adult donors (Supplementary Fig. 9 and Supplementary Data 6). We confirmed three R→G sites in SON DNA binding protein (*SON*), *GRIA3*, and *GRIA4*, which were more highly edited in neurons relative to OLIG. We also validated one S→G site in *UNC80*, a component of the NALCN sodium channel complex (*UNC80*), which was more highly edited in MGE-GABA neurons.

Further, we explored whether our data could resolve the cellular specificity of editing sites previously described to be

dysregulated in bulk brain tissue across neurodevelopment and in neurological disorders (Fig. 3e and Supplementary Data 4). We observed a strong enrichment of MGE-GABA and GLU sites among disease-linked sites, most notably in the dorsolateral prefrontal cortex (DLPFC) and anterior cingulate cortex (ACC) of schizophrenia patients ($p = 0.009$ and $p = 6.3 \times 10^{-5}$, respectively). Also, MGE-GABA and GLU sites were enriched for sites in the DLPFC previously found to be dynamically regulated throughout prenatal and postnatal development[16] ($p = 2.3 \times 10^{-214}$). Enrichment for OLIG RNA editing sites was consistently lower across all independent studies and cohorts, as expected. These results reconfirm the validity of our approach and shed light on some of the cellular origins of altered RNA editing in neurodevelopment and disease[17–22].

**Genes enriched for RNA editing sites within cellular populations**. We found an association between gene length and the number of RNA editing sites per gene within MGE-GABA ($R^2 = 0.16$), GLU ($R^2 = 0.15$), and OLIG cells ($R^2 = 0.04$) (Supplementary Fig. 10A and Supplementary Data 7). After normalizing the total number of RNA editing sites by gene length (*see Methods*), we found a higher density of RNA editing sites in OLIG cells at several genes associated with OLIG-specific expression: *PCDH9* ($n = 560$ sites), *RNF220* ($n = 419$ sites), *FRMD5* ($n = 277$ sites) (Fig. 3f and Supplementary Fig. 10B, C). In MGE-GABA, genes *RBFOX1* ($n = 1696$ sites), *SNHG14* ($n = 1083$ sites), and *DPP10* ($n = 243$ sites) were enriched for editing sites, while in GLU, genes *KCNIP4* ($n = 1831$ sites), *PHACTR1* ($n = 433$ sites), and *PTPRD* ($n = 488$) were enriched, among others (Fig. 3f, Supplementary Fig. 10B, C, and Supplementary Data 7). Notably, genes with higher rates of editing in MGE-GABA and GLU cells were not associated with cell-specific differential gene expression.

**Validation of RNA editing sites by independent snRNA-seq data**. We next examined RNA editing across all brain cell types using single-nuclei RNA-sequencing (snRNA-seq) of the adult PFC generated by PsychENCODE[35] ($n = 3$ independent biological replicates). A total of 24 discrete cell clusters comprising six major cell types were identified through unsupervised dimension reduction and annotated using previously defined cell marker genes (Fig. 4a). To overcome data sparsity associated with snRNA-seq data, we binned nuclei into pseudo-bulk pools that most closely reflect the MGE-GABA, GLU, and OLIG populations in our FANS datasets based on the expression of their markers *SOX6*, *RBFOX3*, and/or *SOX10*, respectively (Fig. 4b and Supplementary Fig. 11). We observed that ~52% of all nuclei expressed GLU markers ($n^{\text{nuclei}} = 8957$), ~21% expressed MGE-GABA markers ($n^{\text{nuclei}} = 3708$), and ~9% expressed OLIG markers ($n^{\text{nuclei}} = 1709$). Notably, a small subset which were assigned to the GLU pseudo-bulk pool were also positive for markers of CGE-derived inhibitory neurons (e.g., *VIP* and *LAMP5*), which make up ~7% of nuclei overall. This was consistent with our FANS strategy, which separated MGE-GABA form other neurons using the MGE-GABA specific marker *SOX6*. Thus, a small population of non-MGE-derived GABA neurons (~10–12% of all GABA neurons) was sorted together with GLU neurons[29]. We generated three additional pseudo-bulk pools representing astrocytes ($n^{\text{nuclei}} = 2,078$), endothelial cells ($n^{\text{nuclei}} = 464$), and microglia ($n^{\text{nuclei}} = 130$). We examined ADAR expression and global editing rates within each of the six cellular pools (see Methods), confirming higher global editing activity as well as higher expression of *ADAR1* and *ADAR2* in MGE-GABA and GLU relative to OLIG cells ($p = 0.002$, linear regression), and

relative to all remaining non-neuronal cell types ($p = 0.0002$) (Fig. 4c).

Furthermore, we queried all selective RNA editing sites derived by FANS (Fig. 2) within each snRNA-seq cellular pool and validated 11,509 sites in MGE-GABA (~10%), 27,209 sites in GLU (~24%), and 3,014 sites in OLIG (~4%) pools with high-confidence (Supplementary Data 8). Of these sites, a total of 1902, 3378, and 215 sites were not cataloged in current RNA editing databases across MGE-GABA, GLU, and OLIG populations, respectively. Sites with validation by snRNA-seq were biased towards the 3′ end of the transcript, and those with higher supporting read coverage displayed a stronger 3′ bias relative to those with lower coverage (Supplementary Fig. 12 and Supplementary Data 8). Notably, the validation rate per cell population functioned as a measure of snRNA-seq coverage thresholds and the number of nuclei included within each pseudo-bulk pool; larger pools had higher validation rates (e.g., the pseudo-bulk GLU pool contained the most nuclei and displayed the highest validate rate) (Fig. 4d). Nevertheless, for the sites passing the defined coverage thresholds, we observed a high level of concordance between editing rates quantified in purified nuclei via FANS relative to editing rates quantified via snRNA-seq for MGE-GABA ($R^2 = 0.54$), GLU ($R^2 = 0.59$), and OLIG ($R^2 = 0.43$) (Fig. 4e). RNA editing sites that did not validate in snRNA-seq were largely explained by the lack of supporting read snRNA-seq coverage or lower expression than observed in FANS (Fig. 4f).

**Imputing cell type-specific RNA editing rates in bulk RNA-seq**. The challenges of cell type purification and single cell sequencing have precluded the broad application of those techniques. To leverage the extensive bulk RNA-seq datasets generated from human brains, we sought to use our data to infer cell type-specific RNA editing in 1129 bulk RNA-seq samples across 13 brain regions from the Genotype-Tissue Expression (GTEx) project (Supplementary Data 9). Given that global editing activity is highest in neurons, we anticipated that a substantial fraction of the variation in global selective A-to-G editing rates in bulk brain tissue would be explained by the proportions of neurons within each sample. Moreover, given the high proportion of neurons in the cerebellum, we expected that editing rates would be highest in this region relative to all others. We used cell type deconvolution to estimate the proportions of six major CNS cell types for each bulk RNA-seq sample (see Methods).

We observed considerable brain region variability in global selective editing activity, defined by the AEI. The cerebellum and cerebellar hemisphere had higher editing activity than all other brain regions ($p = 2.7 \times 10^{-110}$, Cohen's d = 3.00) (Fig. 5a), on par with previous AEI measurements[36]. Similarly, cellular deconvolution of all bulk tissue samples confirmed elevated neuronal proportions in the cerebellum and cerebellar hemisphere relative to all other regions ($p = 1.3 \times 10^{-132}$, Cohen's d = 5.65). Importantly, a significant fraction of the variance in global editing activity was explained by differences in neuronal fractions both within and across all brain regions ($R^2 = 0.31$; Fig. 5b). Donors and regions with higher proportions of neurons displayed elevated global editing activity. Notably, other biological and technical factors were unable to explain as much variation (Supplementary Fig. 13). Using these bulk RNA-seq data, we also confirmed the positive association between the proportion of neurons and *ADAR1* ($R^2 = 0.55$) and *ADAR2* expression ($R^2 = 0.70$), and a negative association with *ADAR3* ($R^2 = 0.22$) (Fig. 5c). For further context, we performed PCA on *ADAR1*, *ADAR2*, and the AEI metric, which accurately stratified all samples by brain region and the proportion of neurons (Supplementary Fig. 14). Collectively, these results confirm the

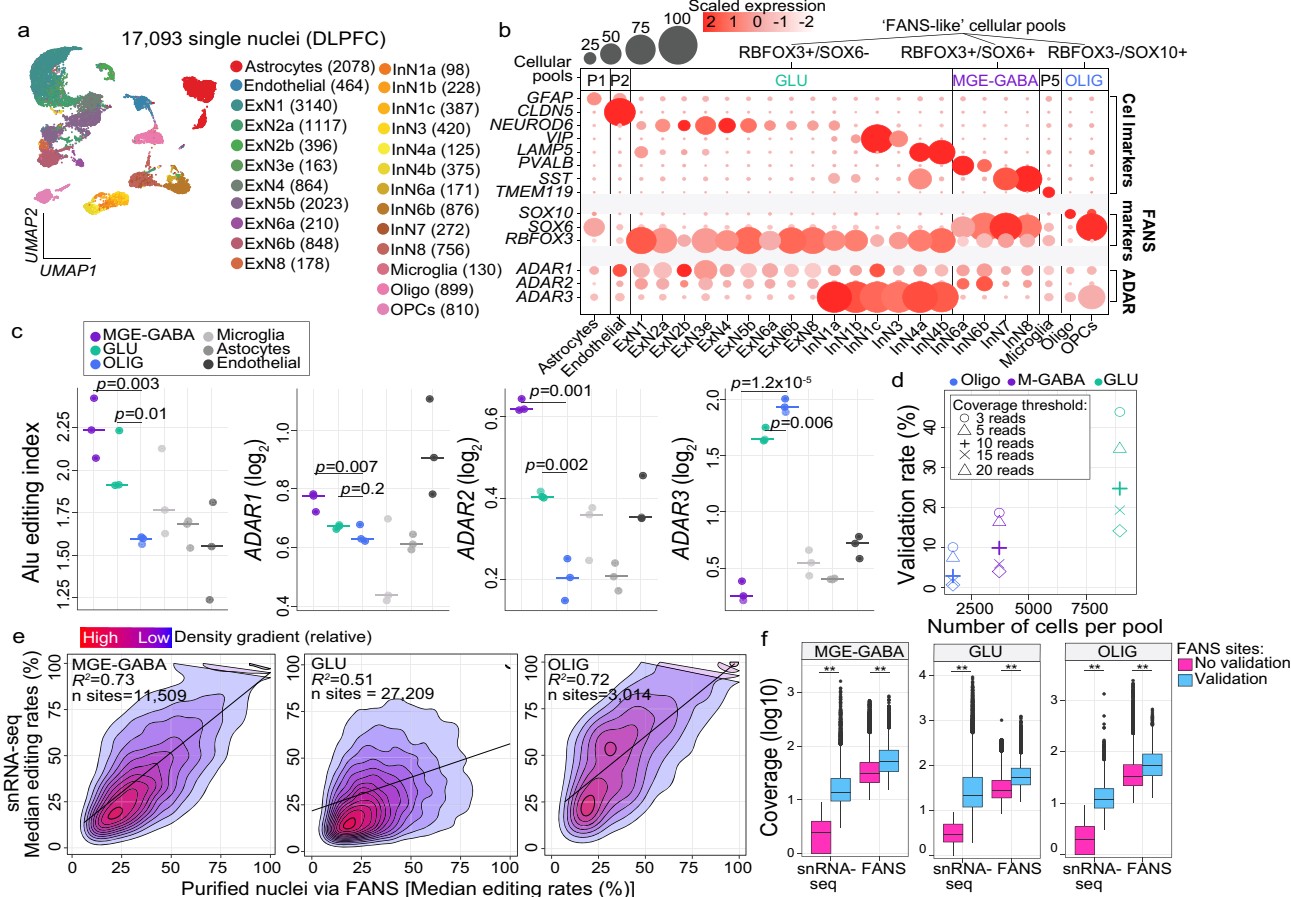

**Fig. 4 Confirmation of cell-specific editing in snRNA-seq. a** snRNA-seq performed on adult PFC ($n = 3$ biologically independent samples) classified 24 unique cell populations. Values in brackets indicate the number of cells per sub-population, including all excitatory (ExN) and inhibitory (InN) neuronal subsets. **b** Cells were collapsed into six major cellular pools (P) and evaluated for expression of canonical marker genes and FANS-derived cell populations (*RBFOX3*, *SOX6*, and/or *SOX10*), which were used to define 'FANS-like cellular pools'. **c** The AEI and expression profiles for *ADAR1*, *ADAR2*, and *ADAR3* (y-axes) were computed for each 'FANS-like cellular pool' in addition to microglia, astrocytes, and endothelial cells. Two-sided regression analyses were used to compute significance between MGE-GABA vs. GLU and GLU vs. OLIG populations and associations with $p < 0.05$ were deemed significant. Data were presented as mean values ± SEM. No adjustments were made for multiple comparisons. **d** The percentage of editing sites (y-axis) from the original FANS-derived cell populations ($n = 9$ biologically independent samples) that validate in each respective 'FANS-like pool' was determined across varying read coverage thresholds. **e** Concordance of median editing levels (%) within each cell type for sites validated by FANS and snRNA-seq. **f** Sites which failed to validate by snRNA-seq were explained by a difference in read coverage and the total number of cells sequenced. All box plots show the medians (horizontal lines), upper and lower quartiles (inner box edges), and 1.5 × the interquartile range (whiskers). Two-sided Mann–Whitney U-tests were used computed significance in mean differences in read coverage differences between sites with and without validation and associations with a $p$ value <0.05 (denoted with double asterisks) were deemed significant. Exact $p$ values are provided in Supplementary Data 8.

cellular specificity of ADARs and the AEI in bulk tissue as observed in our FANS-derived nuclei.

Next, we queried selective editing sites in all GTEx bulk brain samples based on a list of known sites including our 189,229 bona fide cell type-associated sites plus all other sites listed in the REDIportal database (*see Methods*). As expected, the number of editing sites in the cerebellum and cerebellar hemisphere (~58,143 per donor) was threefold greater than all other regions (~18,456 sites per donor, $p = 1.3 \times 10^{-132}$, Cohen's d = 2.99) (Fig. 6a). Intra-donor variation in site detection was similarly associated with the proportion of neurons ($R^2 = 0.70$), *ADAR2* ($R^2 = 0.48$), and *ADAR1* ($R^2 = 0.30$) expression (Supplementary Fig. 15). RNA editing sites mapped primarily to 3′UTRs (~40%) and introns (~22%) with few recoding events (~2.9%) across regions (Supplementary Fig. 16A). Notably, ~20% of all detected sites per donor were cataloged as either cell type-specific or -enriched in either MGE-GABA, GLU, and/or OLIG cells, and these sites displayed significantly higher detection rates in the

cortical regions relative to all other regions ($p = 2.8 \times 10^{-28}$, linear regression) (Fig. 6b). Importantly, these sites also had twentyfold higher editing rates relative to detected sites that were cataloged in the REDIportal database but not by FANS (Fig. 6c). Moreover, not in catalog sites uncovered by FANS were detected at an expected low rate in bulk tissue (~213 sites on average across all donors and GTEx regions) (Supplementary Fig. 16B).

Given the substantial inter-donor variability in the number of detected selective editing sites (Fig. 6a and Supplementary Fig. 14), we asked whether cell type-specific and -enriched sites were consistently detected and edited across the majority of samples (Fig. 6d). The number of commonly edited sites substantially decreases when gradually increasing the requirement of a site to be detected in a larger fraction of donors per region in an incremental fashion (Fig. 6d). Notably, following each iteration, the proportion of retained sites that identify as cell type-associated RNA editing sites gradually increased (Fig. 6e), indicating that such sites are detectable within bulk RNA-seq

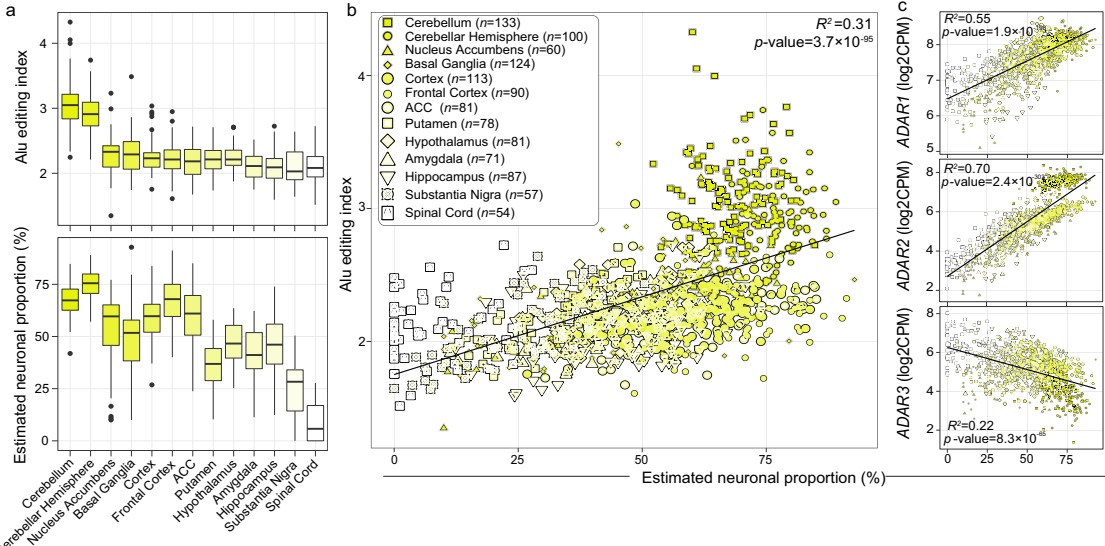

**Fig. 5 Variance in global editing activity in bulk tissue explained by neuronal content. a** *Alu* editing index (AEI) (y-axis, upper) and estimated neuronal cell type proportions (y-axis, lower) across 13 brain regions from the GTEx project (x-axis). Brain regions are ranked by the median level of the AEI. Cellular deconvolution was applied using the Darmanis et al., 2015 reference signature matrix. Box plots show the medians (horizontal lines), upper and lower quartiles (inner box edges), and 1.5 × the interquartile range (whiskers). **b** The amount of AEI variance (y-axis) and T. **c** *ADAR* expression explained by neuronal proportions per region was determined using a regression model across all 13 brain tissues (*ADAR1* upper, *ADAR2*, middle, *ADAR3* lower panel). Associations with a *p* value <0.05 were deemed significant. *R*-squared values are reported and no adjustments were made for multiple comparisons. The total number of independent biological replicates per region is depicted within panel b inset and also provided in Supplementary Data 9.

tissues across the vast majority of donors. We tested this result by computing fold enrichment for all cell type-associated sites relative to sites assigned to specific genic regions and found that these editing sites were disproportionally enriched across the majority of samples ($p = 6.8 \times 10^{-6}$), followed by RNA editing sites mapping to downstream transcription start site positions, sites in exons and those in splice-site regions (Fig. 6f and Supplementary Data 10). These results suggest that sites with cellular resolution comprise those that are most commonly detected across the majority of donors and regions in bulk tissue.

**Genetic variants affect the rate of selective RNA editing.** We used imputed genotype data from the GTEx project to detect common single nucleotide polymorphisms (SNPs) that are associated with RNA editing levels (edQTL, editing quantitative trait loci) (see Methods). To identify genetic variants that could explain the variability of selective RNA editing, we ran association tests across each brain region and identified 661,791 cis-edQTLs (i.e., SNPs located within 1 Mbp of an RNA editing site) at a genome-wide FDR <5% (Fig. 7a). Each max-edQTL (defined as the most significant SNP-site pair per site, if any) meeting a genome-wide significance (FDR < 0.05) was located close to their associated editing site and acting in cis (±200 kb an editing site) (Fig. 7b). Max-edQTLs were examined for cellular specificity according to the aforementioned analyses. A total of 5011, 4514, and 3677 cis-edQTLs were annotated as either MGE-GABA, GLU, and/or OLIG associated, respectively. We found hundreds of cis-edQTLs in each brain region, with an especially large number in the cerebellar regions (Fig. 7c). Overall, a total of 13,438 unique editing sites (eSites) displayed edQTLs and these sites were predominately located in 3′UTRs (~38%) (Fig. 7c). Of these, 1869 eSites (~13.9%) were genetically regulated across three or more brain regions, while 51 eSites corresponding to 47 unique genes were commonly regulated across all thirteen regions (Fig. 7d and Supplementary Data 11). For example, an RNA editing site located on the 3′UTR of glutathione-disulfide

reductase (*GSR*), a site which shows strong preferential editing in neurons, is consistently associated with the same SNP (chr8:30536581) across all 13 brain regions with the similar direction of effect (Fig. 7e). Importantly, our eSites enriched for 618 out of 977 eSites previously identified across GTEx brain regions[27] ($p = 2.1 \times 10^{-31}$, Fisher's exact test) and expand upon these efforts by 13-fold (Supplementary Data 11).

We also compared the absolute effect sizes across all max-edQTLs and observed significantly stronger associations for cell type-specific max-edQTLs relative to those with RNA editing sites that were detected in REDIportal but not by FANS (Fig. 7f). These results were consistent across brain regions (Supplementary Fig. 17). Notably, max-edQTL SNPs were moderately enriched in gene enhancers and promoters that are specific to the brain tested from the FANTOM project collected from SlideBase database[37] (Supplementary Data 11). Additionally, while the sample size is the gold-standard metric to determine power and discovery for QTL studies (Supplementary Fig. 18), the number of unique eSites detected per brain region also correlated with *ADAR1* ($R^2 = 0.59$) and *ADAR2* ($R^2 = 0.85$) expression, as well as with the proportion of neurons per brain region ($R^2 = 0.41$) (Fig. 7g). We confirm these associations using a smaller set of results from an independent analysis of eSite discovery across GTEx brain regions[27] (Supplementary Fig. 19). These results suggest that, in addition to analyzing large sample numbers, edQTL discovery is highly context-dependent: tissues with higher levels of ADAR expression and increased neuronal content provide favorable frameworks for discovery.

## Discussion

Exposing highly regulated editing sites across different cell populations, brain regions and those that are genetically regulated can promote studies to dissect their functional relevance within the right cellular context. Given the dearth of studies examining cellular features of RNA editing in the brain, we first set out to increase the resolution of RNA editing among three main cell

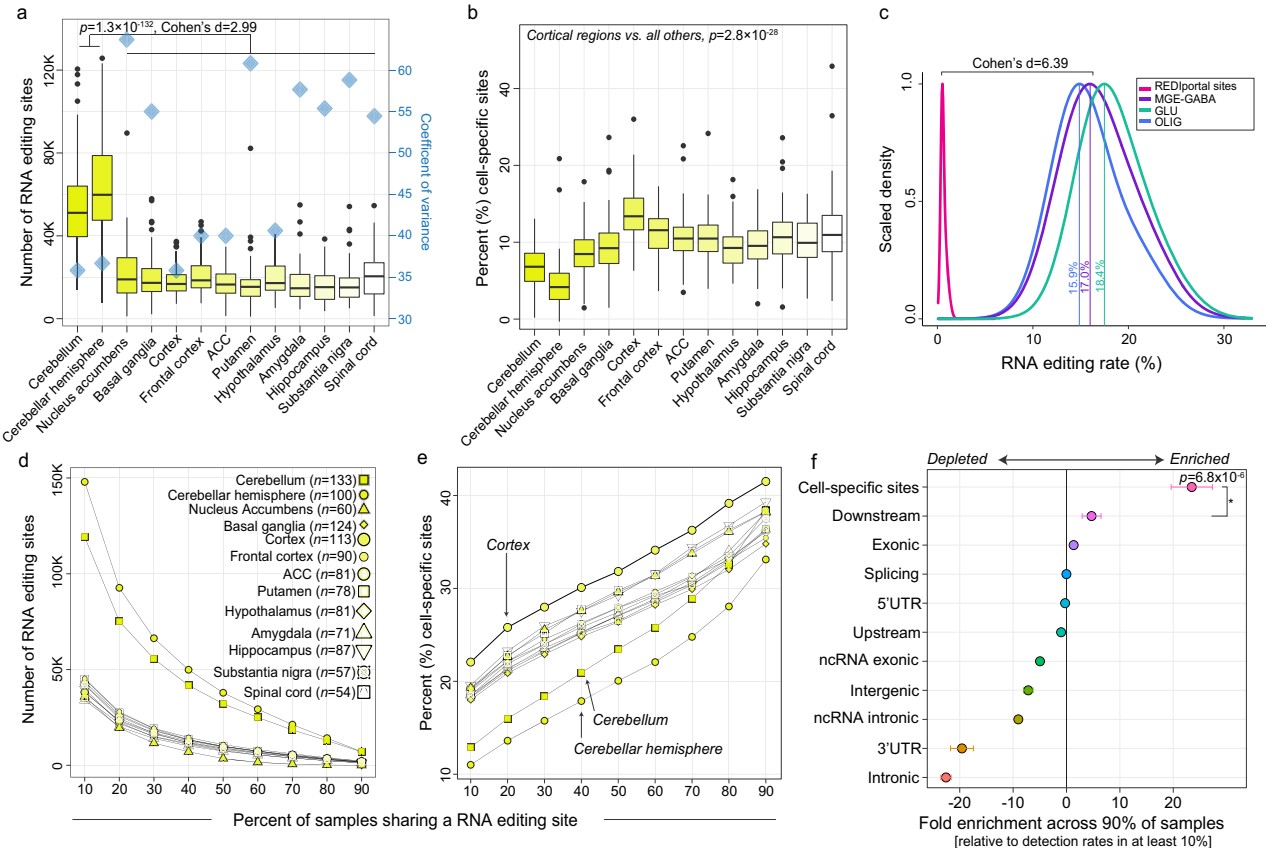

**Fig. 6 Annotation of cell type-associated sites in bulk tissue. a** Total number of selective editing sites (y-axis) reported along with the coefficient of variance (blue diamonds) for each GTEx brain region to describe intraregional variation amongst donors. **b** The percentage of cell type-associated editing sites out of the total number of editing sites per donor (y-axis). Box plots show the medians (horizontal lines), upper and lower quartiles (inner box edges), and 1.5 × the interquartile range (whiskers). **c** Density distribution plots of cell type-associated editing levels relative to all other detected sites in the REDIportal database without cellular annotations. For all tests, a two-sided regression analysis was used to compute significance. No adjustments were made for multiple comparisons. **d** A sliding threshold was used to identify sites shared amongst a given percentage of samples per region (x-axis). **e** Following each threshold, the fraction of cell type-associated editing sites were computed, further highlighting cortical enrichment. **f** Fold-enrichment analysis (sites detected in 90% of samples relative to sites detected in 10% of samples) informs cell type-associated sites are detected across most independent biological replicates (* indicates $p = 6.8 \times 10^{-6}$). Standard error bars reflect brain region level variability. A two-sided regression analysis was used to compute significance. No adjustments were made for multiple comparisons.

populations in the human cortex. Through FANS, unique populations of cell types were isolated from the PFC with distinct functional differences. Using RNA-seq, we determined how RNA editing facilitates transcriptomic diversity more commonly in MGE-GABA and GLU neuronal populations relative to the major glial population in the brain—OLIG cells, documenting numerous sites and genes with cell-specific editing properties. snRNA-seq data was used to validate global trends in editing among CNS cell types and confirm the cellular resolution for a subset of RNA editing sites and genes. Subsequently, we applied rules of cell-specific RNA editing to quantify variability in RNA editing rates observed in bulk brain tissues from the GTEx project, and successfully show that: (i) tissues and donors with higher neuronal content display increased global editing rates and exhibit increased detection rates of RNA editing sites; (ii) commonly detected sites quantified in bulk brain tissue can be classified as cell type-specific or -enriched RNA editing sites; (iii) hundreds of edQTLs classify as cell type-specific or -enriched and display larger effect sizes than edQTLs without cellular resolution; and (iv) similar to RNA editing site detection, edQTL detection is also better powered in regions that express higher levels of *ADAR2* and display increased neuronal content. These results illuminate differences in RNA editing sites and rates among cortical cell

populations and establish frameworks for future large-scale studies of bulk RNA-seq brain tissue to interpret RNA editing sites and their variability.

We observed increased global editing rates in neurons relative to OLIG populations in humans, similar to previous reports among CNS cell types in mice and *Drosophila*[23–25]. These global trends were accompanied by increased detection in the number of RNA-edited sites in neurons. Such differences were largely associated with differences in ADAR expression levels, rather than being driven by differential expression of the edited transcripts. We also uncovered several editing sites in close proximity on the same transcript and co-regulated in the same cell population (e.g., editing sites on *CSMD1* in OLIG, *KCNIP4* in GLU, and *RBFOX1* in MGE-GABA), groups of sites which illustrate a regulation of editing that can exert its effect differently in different parts of the same transcript, as recently shown across neuronal populations in *Drosophila*[25]. While such events would be successively edited by ADAR, regulation by RBPs may also have a similar effect[38]. Several RBPs were preferentially expressed in either MGE-GABA and GLU populations or OLIG and were either positively or negatively correlated with global editing activity. Such RBPs might explain a fraction of the differential RNA editing levels observed at different sites in distinct

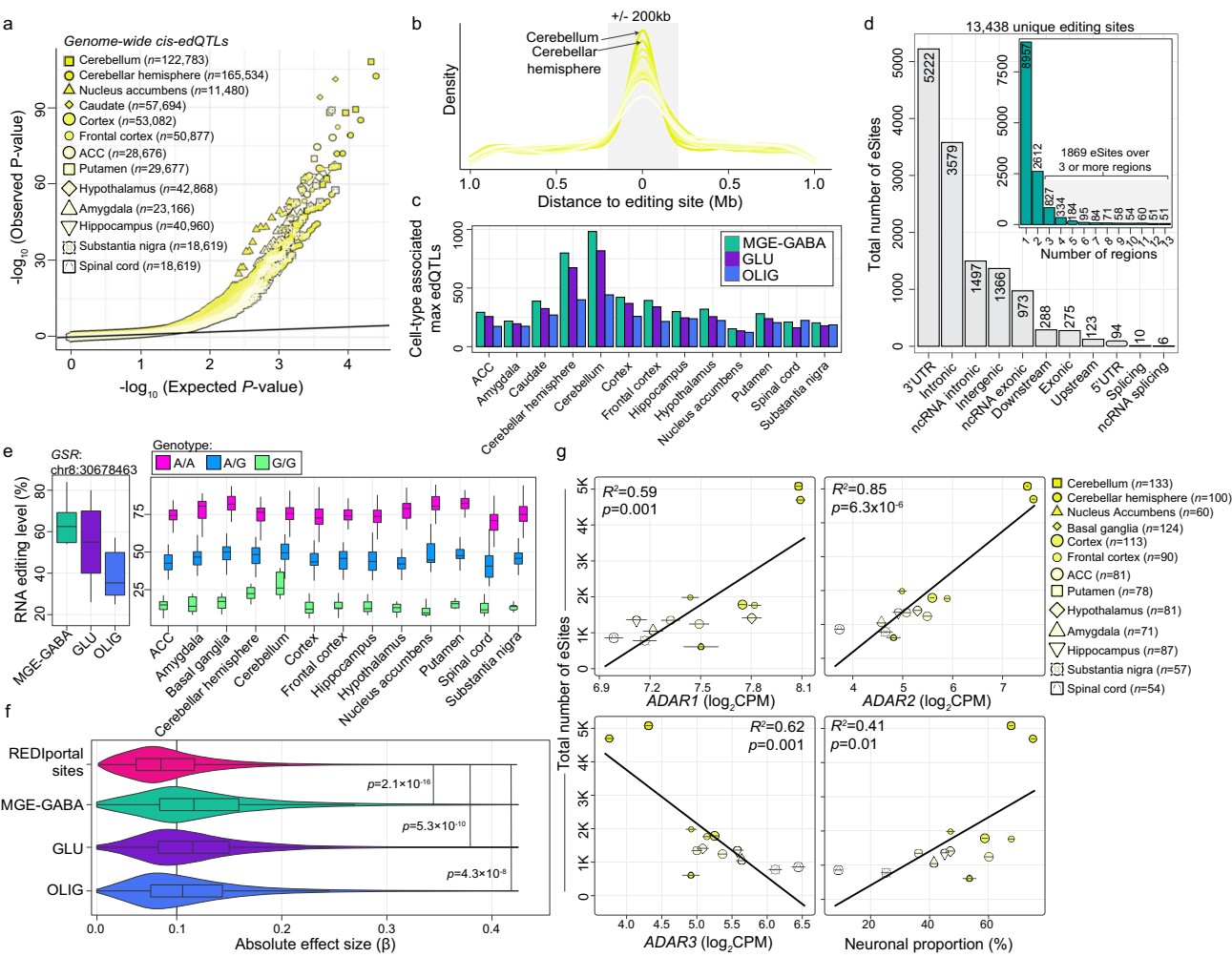

**Fig. 7 Identification and annotation of edQTLs across multiple brain regions. a** QQ-plot depiction of observed versus expected *P* values for genome-wide cis-edQTLs**. b** Max-edQTL SNPs are highly enriched within 200 kb of their respective editing sites. **c** Counts of cell type-associated max-edQTLs in each brain region. **d** The total number of eSites (unique editing sites with associated edQTL) are broken down by genic region. Inset plot counts the number of eSites commonly detected across more than one region. **e** An example of one commonly detected edQTL across all regions with neuronal cell type-enriched editing rates (left) with the consistent direction of edQTL effect across all regions (right). **f** The absolute effect size (β, x-axis) is compared for edQTLs annotated as cell type-associated or edQTLs without cellular identify. All box plots show the medians (horizontal lines), upper and lower quartiles (inner box edges), and 1.5 × the interquartile range (whiskers). **g** Two-sided regression analysis indicates that eSite discovery is dependent upon regions expressing higher levels of *ADAR1, ADAR2,* and those with higher proportions of neurons. No adjustments were made for multiple comparisons. The total number of biological replicates for each GTEx region are provided in panel g and Supplementary Table 9.

populations. However, follow-up functional validation is required to fully understand the *trans* regulation of editing levels by RBPs among brain cell types.

Our study not only provides cellular context for already existing sites in the brain but also identified several previously unknown editing sites likely masked by bulk RNA-seq sampling techniques. For example, a known recoding site (S→G) in *UNC80*, a gene encoding for the NALCN channel complex subunit, was preferentially edited in MGE-GABA neurons (Fig. 3d and Supplementary Fig. 9). This gene is essential for the sensitivity of the sodium leak channel NALCN to extracellular Ca²⁺ and mutations in *UNC80* are associated with congenital infantile encephalopathy, intellectual disability, and growth issues[32]. Moreover, sites not currently cataloged in existing RNA editing databases, which were often found in moderate-to-lowly expressed genes, mostly overlapped *Alu* elements of the transcriptome, consistent with existing reports[3,4,39,40]. Further, our nuclei data, perhaps unsurprisingly, indicate that ADAR-driven RNA editing activity occurs predominately in non-coding transcripts, also

consistent with existing reports[3,4,40]. Such editing sites may regulate circular RNA biogenesis[41] or RNA interference pathways[42], which in turn can alter heterochromatin formation. While follow-up investigations are required to elucidate the functional relevance of these editing sites, these data indicate that transcriptome sequencing of cell populations purified by FANS can accelerate the discovery of editing events, including sites on difficult to detect transcripts.

snRNA-sequencing data were used to confirm global RNA editing trends among MGE-GABA, GLU and OLIG populations, and shed additional light on global editing patterns across three additional CNS cell types. We also validated several thousand cell-specific RNA editing sites using an in silico cellular pooling technique to overcome hurdles related to snRNA-seq. Specifically, calling RNA editing sites from snRNA-seq is challenging as the technique is limited by extremely low capture efficiency and low sequencing depth with reads covering only a fraction of the entire genome[43,44]. Moreover, intronic regions, which were highly edited in FANS-derived data, are often over-represented in RNA-seq

and snRNA-seq generated from the nuclear fraction relative to a combined cytoplasmic and nuclear fraction as in bulk tissue sample and has been demonstrated across various tissues and model systems[45–48]. Validation rates by snRNA-seq were highest for GLU and MGE-GABA populations, which constituted pools with the largest number of nuclei, and concordance of editing levels for these sites was exceptionally high between snRNA-seq and FANS-derived datasets. Until now, there has been limited investigation of the accuracy of RNA editing calls from snRNA-seq data[26]. We anticipate that future work applying statistical models to snRNA-seq to quantify editing will benefit from increased sample sizes (i.e., more biological replicates) and longer, more deeply sequenced reads, which may also expand the catalog of RNA editing sites necessary to build full cell-specific profiles.

Our results also underscore several important features of RNA editing in bulk brain RNA-seq, and in doing so, highlight the tremendous heterogeneity of RNA editing site quantification and detection at the population scale. The majority of randomly selected sites are detected only across a subset of donors within a particular brain region, and these detection rates are largely a function of *ADAR2* expression and whether a region exhibits increased neuronal content (similar to variation in the AEI in bulk tissue). Differences in pools of cellular RNA across samples and randomly sequenced RNA fragments offer likely explanations for such intra-donor variability in site detection. Another possible explanation to account for such heterogeneity is that RNA editing is a transient process and such differences may simply reflect distinctions in the timing of cellular processes and signaling cascades. Indeed, at the level of individual cells, dynamic responses to environmental cues occur on a timescale that is faster than activity-induced transcription via the coordinated, activity-induced switching of internal molecular states and cellular metabolism[1,2]. Moreover, we detected 13,438 unique eSites, which occurred predominantly in 3'UTRs across brain regions. While this level of discovery significantly expands upon existing counts of genetically regulated editing sites in the brain[27], we anticipate that this number is still an underestimate. Our edQTLs do not include rare RNA editing sites that were detected in a small subset of samples (which are the majority) that may also be genetically regulated. Future work considering edQTL models that account for rare editing sites may lead to a more resolved atlas of RNA editing sites and their genetic regulation in the brain.

Finally, while our data offer avenues for understanding cellular specificity of RNA editing, one key question remains: "What is the biological explanation as to why neurons exhibit a preponderance of RNA editing activity?". Here we extend some putative solutions. One possible explanation for enhanced RNA sequence diversity and ADAR expression in neurons might be to afford increased neural plasticity. Neurons, more than other cell types, must be able to quickly respond to altered environmental inputs, therefore, RNA editing may represent a mechanism capable of aligning with the timing required for experience-dependent plasticity, similar to other RNA modifications[49,50]. As such, editing activity in neurons might play a role in controlling the organizational architecture of neuronal networks implicated in higher-order learning and memory. A related explanation of enhanced diversity of RNA editing in neurons may hint at its potential relation with essential neuronal functions. This idea is reinforced by the high density of editing in neuronal transcripts that encode proteins directly involved in excitatory/inhibitory (E/I) functions and the significant differences in editing between excitatory GLU and inhibitory GABA neurons. Such differences may modulate E/I balance within the cortical circuitry. The imbalance of E/I activity is thought to play a critical role in the pathophysiology of several different neuropsychiatric and neurological disorders, including autism spectrum disorder, schizophrenia, and epilepsy[51,52]. Thus, the functional relevance of the observed RNA editing differences between different populations of brain cells in health and disease warrants further investigation. Lastly, it remains unclear how RNA editing may play out across distinct cellular compartments and locations. Recent work has found discrete gene expression differences across synapses, dendrites, axons, and neuronal bodies[53,54], and RNA editing may represent a driver of activity-induced RNA localization[1,2,11].

## Methods

**Experimental design and RNA-sequencing samples**. The following RNA-seq datasets were leveraged to quantify RNA editing. All data is de-identified. Raw FASTQ files or mapped bam files of the following datasets were downloaded from either the NCBI sequencing read archive or Synapse.org:

1. FANS-derived cortical cell populations: Raw FASTQ files were obtained for 27 paired-end (125 bp) nuclei from MGE-GABA ($n = 9$), GLU ($n = 9$), and OLIG ($n = 9$) populations (syn12034263). Antibodies against brain cell population markers NeuN, SOX6, and SOX10 were used in the FANS protocol[34]. In brief, NeuN (also known as RNA-binding protein RBFOX3) is a well-established marker of neuronal nuclei and was used to isolate neuronal (NeuN+) from non-neuronal nuclei (NeuN−); SOX6 is a transcription factor expressed in MGE-GABA neurons during development and into adulthood, and anti-SOX6 antibodies are used to separate nuclei of MGE-GABA (SOX6+) from GLU (SOX6−) neurons[34,55]; finally, SOX10 is a transcription factor specifically expressed in OLIG and is used to isolate OLIG (SOX10+) from other non-neuronal nuclei.
2. Single-nuclei RNA-sequencing: Mapped bam files were obtained for snRNA-seq generated from 17,093 nuclei from the dorsolateral prefrontal cortex (DLPFC) of three adult brains (syn15672826). The 10X Genomics chromium platform was used to capture and barcode single nuclei using the Chromium Single Cell 3′ Library and Gel Bead Kit v2 (10x Genomics) and the Chromium Single Cell A Chip Kit (10x Genomics)[35].
3. Transcriptome and genotype data from Genotype-Tissue Expression Project: We obtained approval to access the Genotype-Tissue Expression (GTEx) Project through the database of Genotypes and Phenotypes (dbGaP) (phs000424.v8). Raw FASTQ files were obtained for a total of 1431 paired-end (75 bp) samples across 13 brain regions (anterior cingulate cortex (ACC), amygdala, caudate, cerebellar hemisphere, cerebellum, cortex, frontal cortex, hippocampus, nucleus accumbens, putamen, spinal cord, substantia nigra). The VCFs for the imputed array data were available through dbGAP, in phg000520.v2.GTEx.MidPoint.Imputation.genotype-calls-vcf.c1.GRU.tar (the archive contains a VCF for chromosomes 1-22 and a VCF for chromosome X).

**Identification of site-selective RNA editing sites**. All FASTQ files were mapped to the human reference genome (GRCh38) using STAR v2.7.3[56] and mapped files were used as input for the subsequent analysis. To quantify site-selective RNA editing sites from FANS-derived cortical cell populations, we used a combination of de novo calling via Reditools v2.0[57] (parameters: -S -s 2 -ss 5 -mrl 50 -q 10 -bq 20 -C -T 2 –os 5) and supervised calling of known RNA editing sites from the REDIportal database[58] as a second pass using samtools mpileup, described below. A number of filtering steps were applied to retain only high-quality, high-confident bona fide RNA editing sites: (i) all multi-allelic events were discarded; (ii) a minimum total read coverage of ten reads and at least three edited reads were required to classify as an editing event; (iii) any sites mapping to homopolymeric regions or in hg38 blacklisted regions of the genome[59] were discarded; (iv) any sites mapping to common genomic variation in dbSNP(v150) and those in gnomAD with minor allele frequency greater than 0.05 were discarded; (v) RNA editing sites within 5 bp of an annotated splice site were removed to avoid issues with mis-mapped reads that should have mapped across splice junctions; and (vi) for FANS-derived cell populations, sites were further filtered using a rate of detection in at least eight out of nine samples per cell type. All remaining sites were annotated using ANNOVAR[60] to gene symbols using RefGene and repeat regions using RepeatMasker v4.1.1[61]. Conservation metrics were gathered from the *phastConsElements30way* table of the UCSC Genome Browser, which consists of evolutionary conservation using *phastCons* and *phyloP* from the PHAST package[62]. Importantly, for sites uniquely detected in one cell type, we performed the third round of RNA editing quantification in the remaining two cell types to determine whether those sites in the neighboring cell types displayed high coverage and little-to-no editing (escaping the minimum coverage and edited read threshold) or simply little-to-no read coverage (escaping the minimum coverage threshold). The resulting RNA editing data frames per cell type contained no more than ~6% missing data. These values were imputed in a cell-specific manner using median imputation (i.e., taking the median editing rate across eight donors per cell type). The resulting sites from these steps were subsequently referred to as high-confidence selective RNA editing sites and were used for downstream analysis.

To quantify RNA editing in snRNA-seq and GTEx data, we called known sites from the REDIportal database in addition to any novel sites (i.e., defined as sites currently not in catalog) identified from FANS-derived cell populations. To this end, to quantify RNA editing sites and rates from a list of given sites, nucleotide coordinates for all such sites were used to extract reads from each sample using the samtools mpileup function[17]. This approach quantifies the total number of edited reads and the total number of unedited reads that map to each RNA editing site detected. All analyses considered read strandedness when appropriate. This supervised analysis required at least ten supporting reads and a minimum of three edited reads to qualify as a RNA editing site.

**Commonly used terms and definitions used in this study**
*Cell type-specific RNA editing site*. Sites uniquely detected in one cell population and with zero coverage (or with insufficient detection rates) in the other two cellular populations.

*Cell type-enriched RNA editing site*. Sites detected in at least two cell populations but with significantly higher RNA editing levels in one of the two cell types, determined by linear regression analysis.

**Computing the *Alu* editing index**. The AEI method v1.0 was leveraged to compute the *Alu* Editing Index for each sample using the STAR mapped bam files as input. The AEI is computed as the ratio of edited reads (A-to-G mismatches) over the total coverage of adenosines and is a robust measure that retains the full *Alu* editing signal, including editing events residing in low-coverage regions with a low false discovery rate.

**Quantification of RNA hyper-editing**. RNA reads that undergo extensive hyper-editing of many neighboring adenosines within an extended region or cluster on the same transcript will not align to the reference genome due to the high degree of dissimilarity. Therefore, to identify hyper-edited reads in the current study, all unmapped reads from the original STAR alignment were converted to FASTQ and used as input for hyper-editing analysis. We adopted a well-established RNA hyper-editing pipeline[7] with minor additional processing steps[15], including: (1) extending cluster boundaries by the average distance between editing sites per cluster and subsequently merging clusters with overlapping coordinates (cluster length is a commonly a product of read length); (2) all resulting hyper-editing sites were annotated using ANNOVAR (described above); (3) sites mapping to common genomic variation in dbSNP(v150) (maf > 0.05) were discarded.

To minimize any batch effects, we computed a normalized hyper-editing signal per million mapped bases[7]. The normalized hyper-editing signal was computed by dividing the total number of resulting high-quality RNA hyper-editing sites over the total number of mapped bases from the STAR alignment and multiplying the resulting value by one million. The number of total uniquely mapped bases for each sample were collected using Picard Tools v2.22.3 (http://broadinstitute.github.io/picard/) on each mapped bam file.

**Local motif enrichment analysis**. EDLogo was used to quantify local sequence motifs[63]. We pulled sequences 4 bp (±) of the target adenosine to evaluate the enrichment and depletion of specific nucleotides neighboring A-to-G editing sites.

**Analysis of RNA editing by gene length**. The total number of selective A-to-G RNA editing events were computed for each gene within each cell population. The total number of edits per gene were compared in a series of pairwise comparisons across each cell populations. To adjust for gene length, we normalized the number of A-to-G editing events per gene for each cell type by the log of gene length (number of edits per gene/$\log_2$(gene length + 1)). Subsequent pairwise comparisons of the normalized number of RNA editing events per gene were performed (MGE-GABA vs. GLU, MGE-GABA vs OLIG, GLU vs. OLIG). Genes displaying enrichment of normalized RNA editing sites were those dented as outlier genes beyond the 99% confidence intervals from the grand mean.

**RNA-binding protein and eCLIP enrichment analysis**. RNA-binding proteins (RBPs) were first defined based on a consensus list of 837 high-confidence human RBPs from the hRBPome database[64]. A total of 470 RBPs were detected in FANS-derive cell populations and subjected to further analysis. Differential expression of these RBPs was computed using a linear model through the limma R package[65] covarying for the possible influence of age and PMI. Donor as a repeated measure was controlled for using the duplicateCorrelation function in limma[65]. A matching approach was used to test for all genes genome-wide for subsequent analyses. To explore putative *trans* regulators of RNA editing, we further studied all RBPs that were significantly differentially expressed. RBPs were further subjected to correlation analysis with the AEI metric using Pearson's correlation coefficient.

Next, we collected transcriptome-wide binding patterns of FMRP (an RBP in the current analysis known to interact with *ADAR*). Data from two eCLIP experiments and an input control experiment were obtained using the postmortem frontal cortex from control subjects[18]. At the time of writing this paper, this experiment represents the only eCLIP-seq data generated from human postmortem brain tissue. The regioneR R package[66] was used to test overlaps of FMRP binding regions for each replicate separately with cell type-specific RNA editing sites based on permutation sampling. We repeatedly sampled random regions from the genome 1000 times, matching the size and chromosomal distribution of the region set under study. By re-computing the overlap with FMRP binding sites in each permutation, the statistical significance of the observed overlap was computed.

**Cell type-enrichment of RNA editing levels by differential editing analysis**. To identify sites with differing levels of RNA editing between two given cell types, we implemented linear model through the *limma* R package[65] covarying for the possible influence of age, PMI, and donor as a repeated measure using the duplicateCorrelation function. Secondary models covaried for *ADAR1* and *ADAR2* expression (the catalytically active editing enzymes) to explore potential ADAR-dependent editing activity. All significance values were adjusted for multiple testing using the Benjamini and Hochberg (BH) method to control the false discovery rate (FDR). Sites passing a multiple test corrected *P* value <0.05 were labeled significant.

**Gene ontology enrichment of differentially edited sites**. Genes harboring differentially edited sites were functionally annotated using SynGO to test for synaptic gene enrichment (https://syngoportal.org/) using default parameters. Further, the ToppFunn module of ToppGene Suite software[67] was also used to examine enriched biological processes. For this analysis, set a genomic background defined as all genes harboring at least one editing site and tested for significance using a one-tailed hypergeometric distribution with a Bonferroni correction. This is a proportion test that assumes a binomial distribution and independence for the probability of any gene belonging to any set. We use a one-sided test because we are explicitly testing for over-representation of genes that harbor editing sites across hundreds of GO categories, without any a priori selection of candidate gene sets. Using this framework, we tested enrichment among (i) all differentially edited sites and (ii) all differentially edited sites with the exception of intronic sites (i.e., removing intronic sites, which were the majority) to gauge their overall influence on enrichment results.

**Enrichment of cell-specific sites across neurodevelopment and neurological disorders**. Both cell-specific and –enriched RNA editing sites were interrogated for over-representation of RNA editing sites previously found to be dysregulated across neurodevelopment and in neuropsychiatric disorders. Transcriptome-derived sets of RNA editing sites were curated based on the following curation of RNA editing sites: two lists of sites found to change in editing rates across prenatal and postnatal cortical development[15,16]; dysregulated RNA editing sites in schizophrenia[17], Fragile X Syndrome, and ASD[18]. To compute the significance of all intersections, we used the GeneOverlap function in R which uses a Fisher's exact test and an estimated odds ratio for all pairwise tests based on a background set of genes detected in the current study.

**Predicting consequence of RNA editing on splicing**. We applied SpliceAI[68] to cell type-associated intronic sites in Supplementary Data 2. SpliceAI is a deep neural network that accurately predicts splice junctions from an arbitrary pre-mRNA transcript sequence. We used a delta score probability of 0.5 or higher as a threshold for an RNA editing site being splice-altering. Notably, we removed sites within 5 bp of a known splice site before running this approach and thus results likely represent a low estimate of truly splice-altering events.

**Validation of recoding sites by targeted PCR amplification and high-throughput sequencing**. We selected four cell type-specific recoding RNA editing sites to validate using an independent approach. We employed brain tissue samples from the orbitofrontal cortex (OFC) in an independent cohort of six adult individuals without a neuropsychiatric diagnosis. Nuclei of MGE-GABA, GLU, and OLIG cells were isolated by FANS, followed by RNA extraction and library construction. PCR primers were designed to target four recoding RNA sites in *UNC80*, *GRIA3*, *GRIA4*, and *SON* (Supplementary Fig. 9). PCR primer sequences are available in Supplementary Data 6. The rhAmpSeq targeted amplicon sequencing kit (Integrated DNA Technologies) was used in two rounds of PCR amplification, to obtain the targeted PCR products (PCR1) and to introduce unique indexes for multiplex sequencing (PCR2), according rhAMPSeq protocol. Three µg of RNA-seq library was used in the first round of PCR for each subject and cell type. The resulting rhAmpSeq libraries were cleaned using Agencourt AMPure XP beads (Beckman Coulter, Inc), pooled, and sequenced on MiSeq obtaining at least 15,000 reads per sample (median read number per sample: 15,700 reads). The sequencing data (FASTQ files) were used to quantify the RNA editing read numbers and editing percentages at the four studied sites by counting reads which mapped identically to the ±10 bases surrounding the edited site.

**snRNA-seq to identify and validate RNA editing sites**. Mapped bam files were obtained following the alignment of short reads to the reference genome (GRCh38/hg38). Reads were filtered for low quality, and counts were quantified (cell barcode counts and unique molecular identifier counts for each annotated gene) using CellRanger count, as previously described[35]. To optimize cell classification and

reduce unwanted variance, we quality filtered, normalized, and scaled data according to Seurat's guidelines. For these data, we used a set of previously implemented methods[35] consisting of the following steps. First, a cell was excluded if the number of expressed genes was less than 300 or more than 7000, with the number of UMI less than 300 or more than 20000, or the percentage of mitochondria reads more than 5%. The normalization method was LogNormalize and the scale factor was 10000. The linear regression was performed by choosing the percentage of mitochondria reads as a variable. Rfrom was used to compute the specificity score for each gene in each cell cluster. Hierarchical clustering was manually checked along with the top-ranked genes in each cell cluster to determine cellular specificity based on well-known gene markers to verify the assignment of cell types and subtypes. Cells with inconsistent or no assignment were removed from the analysis.

Like the original report, we identified 29 transcriptionally distinct cell clusters representing various populations of glutamatergic excitatory projection neurons, GABAergic interneurons, oligodendrocyte progenitor cells, oligodendrocytes, astrocytes, microglia, endothelial cells, and mural cells. Here, we further reduced these clusters into 24 cell clusters by collapsing five highly similar GLU cell subsets based on *NEUROD6* expression. Additionally, we further collapsed these cell clusters into six main cellular pools that constitute MGE-GABA, GLU, and OLIG populations (based on the presence/absence of markers *RBFOX3*, *SOX6*, *SOX10*) and astrocyte (*GFAP*), microglia (*TNEN119*) and endothelial cell (*CLDN5*) populations.

To quantify RNA editing sites from snRNA-seq data, each mapped bam files was parsed into six unique bam files (per donor) reflective of the six main cellular pools (described above) using each cell's unique molecular identifier (UMI). In this way, we pooled cells with matching cell type markers and gene expression patterns for subsequent RNA editing analysis. Next, we quantified RNA editing levels for all high-confidence sites identified in the FANS-derived cell populations using the samtools mpileup function (described above). Given the challenges related to the limited number of cells per pool and low sequencing depth, we explored validation rates under varying read coverage thresholds (minimum of 3–20 reads), but ultimately required minimum coverage of 10 reads per site. We defined sites that were validated as those with median editing rates across the three replicates that were within 50% of the median editing rate of the FANS-derived site.

We tested whether sites that validate based on snRNA-seq were skewed toward the 3′ end of a transcript. First, we used RseqQC (http://rseqc.sourceforge.net/) to compute RNA-seq read coverage over the entire gene body for all transcripts for both the FANS RNA-seq data and snRNA-seq pools. Indexed bam files were used as input and the "wgEncodeGencodeBasicV31" table was downloaded from the UCSC table browser and used as our reference gene model in bed format. Second, for all sites, we computed the distance to the respective transcription start site (TSS). Sites were next parsed based on those that were validated by snRNA-seq and subsequently binned into three groups based on total read coverage: high (first quintile), medium (second, third, fourth quintiles), and low (fifth quintile). A two-sided Mann–Whitney *U*-test was used to test whether sites that validate by snRNA-seq (and each respective coverage bin) was significantly further from the TSS compared to those sites which did not validate.

**GTEx data pre-processing and cell type deconvolution.** All GTEx FASTQ files were mapped to a human reference genome (GRCh38) using STAR v2.7.3[56] and counted using featureCounts[69]. Count matrices were assembled for each brain region, filtered to retain genes with at least 1 count per million in at least half of the samples per region, and VOOM normalized using limma[65]. Each resulting normalized data frame was subjected to principal component analysis to identify and remove any outlier samples that lay beyond two standard deviations from the grand mean. Following the outlier removed, a total of 1129 samples were retained for all subsequent analyses.

To compute cellular composition, we applied non-negative least squares (NNLS) from the bMIND R package[70] and utilized the Darmanis et al., signature matrix[71] which contained a mixture of six major cell types: astrocytes, oligodendrocytes, microglia, endothelial cells, excitatory and inhibitory neurons. NNLS, executed through the est_frac function, was applied to $\log_2$ count per million (CPM) transformed data using the *limma* package in R. For each sample, both excitatory and inhibitory neuronal predictions were summed into one "neuronal" cell population. We focus our predictions on these major cell types in an effort to reduce noise and to evaluate distribution of cell types that reflect an approximate expected distribution in the human brain based on prior work.

**GTEx bulk brain cis-edQTL analysis.** We conducted *cis*-eQTL mapping within the 13 brain regions. We leveraged existing VCFs for high-quality imputed genotype array data from dbGAP (phg000520.v2.GTEx.MidPoint.Imputation.genotype-calls-vcf.c1.GRU.tar), using SNPs with an and estimated minor allele frequency ≥0.05. Only RNA editing sites detected in at least 50% of samples per region were considered for *cis*-edQTL mapping. Each data matrix for each region exhibited ~17% missing values, which were imputed using the well-validated predictive mean matching method in the *mice* R package using five multiple imputations and 30 iterations[72]. The genomic coordinates for all RNA editing sites were lifted over to GRCh37. Subsequently, to map genome-wide edQTLs, a linear model was used on the imputed genotype dosages and RNA editing levels using

MatrixEQTL[73]. RNA editing levels were covaried for sex, age, RIN, and type of death. To control for multiple tests, the FDR was estimated for all *cis*-edQTLs (defined as 1 Mb between SNP marker and editing position), controlling for FDR across all chromosomes. Significant *cis*-edQTLs were identified using a genome-wide significance threshold (FDR <0.05). To assess whether *cis*-edQTLs relate to brain promoter and enhancer regions, the overlap between max *cis*-edQTLs and promoter and enhancer regions was tested from the FANTOM project collected from the SlideBase database[37]. A permutation-based approach with 1000 random permutations was used to determine the statistical significance of the overlap between edSNP coordinates and enhancer regions using the R package regioneR[59].

**Reporting summary**. Further information on research design is available in the Nature Research Reporting Summary linked to this article.

## Data availability

Raw RNA-sequencing FASTQ files across MGE-GABA, GLU, and OLIG cell populations in this study have been deposited in the Synapse database under accession code syn12034263. Cell type-specific RNA editing sites generated in this study are provided in Supplementary Data 2. Raw snRNA-seq files in this study have been deposited in the Synapse database under accession code syn15672826. Approved access can be obtained for GTEx data through dbGaP (phs000424.v8). Further, to promote the exchange of this information, we developed an interactive R Shiny app with an easily searchable interface to act as a companion site for this paper: https://breenms.shinyapps.io/CNS_RNA_Editing/.

## Code availability

All code and summary statistics are provided on GitHub: https://github.com/BreenMS/RNA-editing-in-CNS-cell-types and https://github.com/ryncuddleston/RNA-hyper-editing.

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

## Acknowledgements

We thank the GTEx consortium for making their RNA-sequencing data publicly available. M.S.B. is a Seaver Foundation Faculty Scholar.

## Author contributions

M.S.B. performed the majority of the bioinformatics data analyses, generated all figures and tables, and led data interpretation. W.H.D. performed hyper-editing. J.L. called and annotated RNA editing sites in FANS and snRNA-seq data and performed SpliceAI analysis. X.F. downloaded all data used in the current study, supported QC of all RNA-seq fastq files and imputed genotypes, and developed the R Shiny app as a companion to this paper. S.K. and A.K. performed independent validation of recoding sites. M.L. and E.A.M. aided in design and interpretation. M.S.B. and S.D. conceived the study. M.S.B.,

S.D., and E.A.M. provided funding for the current study. M.S.B designed the analyses and wrote the paper. All authors read, edited, and approved the final manuscript.

## Competing interests

The authors declare no competing interests.
