## [Peer Review File · Nature Communications]

Title: Cellular and genetic drivers of RNA editing variation in the human brainREVIEWER COMMENTS

Reviewer #1 (Remarks to the Author):

In the manuscript, entitled "Cellular and genetic drivers of RNA editing variation in the human brain", Cuddleston et al. investigate A-to-I RNA editing using publicly available RNA-Seq datasets including three major cell populations: MGE-GABA neurons, glutamatergic neurons, and oligodendrocytes. They show that the pattern of RNA editing is highly cell type-specific, with more A-to-I sites detected in neurons relative to oligodendrocytes. Using the imputed SNP data from the GTEx project, the authors further perform association tests and identified a large number of cis-edQTLs. Although the idea and approach are not novel (such as the study from Park et al. *Genome Biology* 2017, for A-to-I RNA editing and edQTL analysis across 445 human LCLs), this manuscript addresses an interesting topic in the field and utilizes a wide range of publicly available datasets. However, I have several concerns regarding the analysis approach and the data presented, which greatly dampens enthusiasm for the manuscript in its current form.

Major concerns:

1. It is well known that Cohen's d is frequently used in estimating sample sizes for statistical testing. Here, it is not clear why the authors applied Cohen's d when comparing AEI, the expression of ADAR genes, RNA hyper-editing sites, etc., among the MGE-GABA, GLU and OLIG groups, and what is the rationale by using Cohen's d when sample size is only 9 per group?

2. I am surprised to see such a big difference in numbers of RNA hyper-editing sites and cell type-specific sites among MGE-GABA, GLU and OLIG. This is interesting. However, the authors didn't present basic reads and mapping stats for three groups of samples. Thus, it is impossible to assess whether sequencing yields and the quality of sequencing reads (e.g. mapping ratio and high quality mapping ratio) biased these numbers.

Notably, sample "H427" in the GLU group also shows a very low number of RNA hyper-editing sites compared to others in the same group. Is there any indication of the reason from this sample? The average number of RNA hyper-editing sites in the GLU group was apparently dragged down by this sample.

It is also worth noting that, in M&M (Line 144), the authors stated that sites should meet the criterion of in at least 8 out of 9 donors, which seems to compromise with this sample in the GLU group, but I am concerned that it may even cause more bias (of the difference) when comparing neuron groups with the OLIG group.

3. In Supplemental Table 1, the normalized HE counts seem very large. For example, the number of HE counts and normalized HE counts for sample "H276" are 282599 and 16.99, respectively. Based on the definition of the "normalized HE counts" mentioned in the Results section ("...we normalized the hyper-editing signal to the number of mapped bases per sample..."), I can calculate the number of mapped bases for this sample is only 16,633 bp, which doesn't make sense at all. The author should correctly

describe how they generated the “normalized HE counts”. Again, the basic reads and mapping stats are essential to understand what was exactly happening here.

4. Lines 130-131, the authors stated that “Hyper-editing sites commonly occurred in introns and 3’UTRs”.

Line 154: “Second, ~73% of editing events were detected in introns, while only a small fraction impacted protein coding regions (~0.67%) (Figure 2D)”.

It is a bit surprising to see very high signals in introns from RNA-Seq data. Are these all related to intron retentions? I assume most RNA-Seq reads should be aligned to the exon regions rather than the introns. However, there are no stats for me to check this information.

In previous studies, such as Park et al (Genome Res. September 2012 22: 1626-1633) showed examples of overhanging RNA-Seq reads that were mapped incorrectly into the intron when the correct position should be in the adjacent exon, even though the splice junction was provided to the read mapper. It is not clear how many intron-derived sites were due to the above-mentioned misalignment.

In addition, it seems that the authors didn’t consider whether consecutive adenosines are more common in introns and 3’UTRs than other regions, which may also cause the bias of what the authors find in this study.

5. The authors used publicly available snRNA-seq data to validate the RNA editing sites. However, the validation rates vary in three groups (from ~3% to ~27%). What does this variance mean? Does it indicate less accurate A-to-I call for the GABA and OLIG groups? Moreover, I think it would be more informative if the authors can extract the number of sites that were not successfully validated but have enough coverage in the snRNA data – for the estimation of false discovery rates.

6.

Lines 567-576

Residual (e.g. the resid function in the R package) is the difference between an observed value and the value predicted by a regression model. It is not clear why & how the authors adjusted the gene length using the resid function. To normalize the data by gene length, a more straightforward approach is to normalize the gene size to a certain length (e.g. 1 kb).

Minor issues:

1. Some mistakes in writing:

i) “..., discovery rates consistent global editing activity (Figure 1)”

ii) “Raw FASTQ files were obtained for a total of 27 paired-end (125 bp) nuclei samples were collected from ...”

iii) “Raw FASTQ files were obtained for a total of 1431 paired-end (75 bp) samples were collected across 13 brain regions...”

- iv) Line 345: what is “200kb ± bp the target adenosine”?
- v) the Figure legend of 7B: “+/-200bp”?

2. Lines 131-133: “... and enriched for a local RNA editing sequence motif whereby guanosine is depleted -1bp upstream and enriched +1bp downstream the target adenosine (Figure S1D-E); validating the accuracy of our hyper-editing approach.”

I don't see the point why the accuracy of our hyper-editing approach was validated here. If it is in line with previous reports, the authors need to correctly cite the relevant publications.

3. Line 146: “107,999” - should it be 107,998 based on Supplemental Table 2?

4. What is the definition of “the 189,229 cell type-associated sites” mentioned in Abstract? It is not mentioned in M&M and Results sections.

5. Line 148: “We defined ~60% of these sites as cell type-specific as they were uniquely detected in one cell type, ...”.

“~60% of these sites as cell type-specific” seems not correct. For example, in Figure 2A the venn diagram, GLU specific sites is obviously less than 40%.

6. In Fig 3B, for the cell type-enriched sites, the intersection of OLIG and GLU has two numbers, 11508 and 24. Please correct.

7. In GO enrichment analysis, it is not clear how the authors selected genes if the differentially edited sites were in intergenic or intron regions. Given these two categories contain most of the RNA editing sites, it will significantly affect the GO results.

8. Regarding the RNA recoding events, Supplemental Table 5 lists 84 sites, while the authors claimed there were 38 RNA recoding events (line 213). This is inconsistent. Please clarify.

9. Line 685: “To control for multiple tests, the FDR was estimated for all cis-edQTLs (defined as 1Mb between SNP marker and editing position), ...”.

So, what is the number of all cis-edQTLs for the multiple tests control?

10. Line 688: what exactly does it mean by “brain promoter and enhancer regions”?

Reviewer #2 (Remarks to the Author):

In this manuscript entitled “Cellular and genetic drivers of RNA editing variation in the human brain”, Cuddleston et al. studied cell type-specific RNA editing sites in human brain. They first called cell type-specific RNA editing sites in three major cell populations including medial ganglionic eminence (MGE) - derived inhibitory GABAergic interneurons (MGE-GABA), excitatory glutamatergic neurons (GLU), and oligodendrocytes (OLIG) from adult prefrontal cortex (PFC) by fluorescence-activated nuclei sorting (FANS). These results showed that neuron-specific editing sites are of higher editing levels than those of OLIG, and cell-specific editing sites are related to different biological pathways. Then they validated these results by single-nuclei RNA sequencing. Next they further verified that RNA editing levels are correlated with neuron proportions from 13 brain regions by using data from GTEx project. Finally, they identified genetic variants associated with RNA editing levels. This work provides lines of evidence that RNA editing levels are correlated with cell types in brain, indicating the potential biological regulation roles of RNA editing in different brain cell types. The manuscript is well organized and written. There are only several minor suggestions for the authors to consider during revision.

1. The authors analyzed RNA editing sites among three major cell populations (MGE-GABA, GLU, and OLIG) purified from adult prefrontal cortex (PFC). It will be much appreciated if the authors could specify why not choose other than PFC regions for the analysis. In addition, since at least 24 unique cell populations were detected in PFC (Figure 4A), the authors may also want to explain a little bit how many of them belong to MGE-GABA or GLU, respectively, or their proportions in PFC.
2. In Line 180-185, about 400 differential expressed RBPs were shown to be correlated with global editing activity, however only FMRP was chosen for further eCLIP validations. Why?
3. In Line 186-190 and Figure S4D-E, the authors got the conclusion that FMRP eCLIP peaks were significantly enriched for MGE-GABA (permutation p-value=0.005, Z-score=9.3) and GLU sites (permutation p-value=0.005, Z-score=6.2), and moderately enriched for OLIG sites (permutation p-value=0.005, Z-score=7.4). As the Z-score of GLU sites is smaller than that of OLIG sites, it is not clear how the authors concluded that GLU sites are more enriched than OLIG sites?
4. In Line 249-250, authors mentioned that “genes with higher rates of editing in MGE-GABA and GLU cells were not associated with cell-specific differential gene expression”. If the editing didn’t influence gene expression, what is the biological functions of editing? Whether splicing or secondary structures were regulated by these neuron-specific editing sites?
5. In Figure 4C, the ADAR1 expression level of endothelial was higher than those in five types of cells including two type of neurons (MGE-GABA and GLU) , but the editing levels in endothelial were not comparable to those in two types of neurons (MGE-GABA and GLU). Any explanation?
6. Generally, snRNA-sequencing uses a 3'-end capture strategy for RNA enrichment and sequencing, and

thus only parts of genes/3'UTR (about 200~500 bp long) could be sequenced. If it were the case, RNA editing calling from snRNA-sequencing might be incomplete/skewed due to the 3' bias. The authors at least need to discuss this point in the manuscript.

7. Some minor changes:

In Figure 3B, the overlap between OLIG and GLU wrongly occurred twice.

The "Alu" should be italic, and please check throughout the manuscript.

In Figure 1C, 5C, 7G, the correlation coefficients (R2) are somehow mistakenly labelled less than 0 (for example, R2=-0.33 in Figure 1C).

In Figure 4A, what are the full names of ExN and InN cells.

In Line 180-185, the name of RBPs such as "NOP14, PTEN and DYNC1H1" shouldn't be italic.

In Line 345, "200kb ± bp the target adenosine" missed the range number. In addition, is 200kb too far away as "cis"?

Reviewer #3 (Remarks to the Author):

The manuscript by Breen et al. has used the FANS and single-nuclei RNA-sequencing ways to study the RND editing sites across the glutamatergic, MGE GABAergic neurons, and the oligodendrocytes cells from the human prefrontal cortex. The authors found 189,229 cell type-associated RNA editing sites and the neurons have a higher number of editing sites than oligodendrocytes. In addition, 661,791 cis-editing quantitative trait loci from various brain regions were discovered. This study provided the human RNA editing sites atlas and showed the cell type-specific editing sites, however, except some novel editing sites were found, no major outstanding discoveries were found in this study. I have some comments that I hope will help further improve this paper.

Major comments:

1. The manuscript mainly presented the discovered RNA editing sites and the sites number comparison between the three types of cell types, however, the findings in this work are not new since the vast majority of RNA editing sites have been detected in the published works, including the finding of editing levels in neurons are higher than glial cells. From my point of view, if the authors can highlight some of the novel found RND editing sites, and explain the internal mechanism of why the sites are higher in neurons relative to oligodendrocytes, and linking the newfound sites to the regulation of neuronal transcription, splicing, and functionality.
2. could you explain why the expression of ADAR1 in GLU cells is significantly higher than MGE-GABA neurons, while the opposites results are found for the ADAR2 between these two cell types in the Figure 1B? which seems are not matchable to Figure 1C. And why the oligodendrocyte cells have the highest expression profiles of ADAR3 than neurons in Figure 1B?
3. In Figure 2H, the authors showed that the found novel A-to-G sites displayed significantly more supporting read coverage compared to known sites, could you give the possible explanations? Did the

authors found the same significance for the novel found C-to-T and G-to-A editing sites?

4. In Figure 3A, the authors performed the differential editing analysis of the cell type-enrichment of RNA editing levels, some differentiated sites were marked on the volcano plots, but I didn't find some text explanation of these differentiated sites, could the authors explain why these sites were marked, did these sites showing some linking to the cell type-specific molecular functionality?

5. The details of the preprocessing, normalization, dimension reduction, and clustering for the single nuclei sequencing data analysis should be included in the method part.

Minor comments:

1. is the full name of ADAR (adenosine acting on RNA) correct? Should it be Adenosine deaminases acting on RNA?

2. statistical analysis details such as method and p-value threshold selection need to be given in the method part.

3. could you show the statistical analysis values in Figure 2I?

We sincerely thank the reviewers for their generous time and careful review of our work. We are also very appreciative of the reviewers' recognition of the novelty and importance of this study and its impact. We provide point-by-point responses to all major and minor critiques below. In addressing each comment, we present a revised body of work that reflects the scientific rigor, clarity, and reproducibility required for publication. For clarity, we provide two versions of our manuscript: 1) a version where text changes are modified by **yellow highlight**; and 2) a modified version without any track changes or highlight. Again, thank you for your time and further consideration of our revised manuscript.

REVIEWER COMMENTS

Reviewer #1 (Remarks to the Author):

In the manuscript, entitled "Cellular and genetic drivers of RNA editing variation in the human brain", Cuddeleston et al. investigate A-to-I RNA editing using publicly available RNA-Seq datasets including three major cell populations: MGE-GABA neurons, glutamatergic neurons, and oligodendrocytes. They show that the pattern of RNA editing is highly cell type-specific, with more A-to-I sites detected in neurons relative to oligodendrocytes. Using the imputed SNP data from the GTEx project, the authors further perform association tests and identified a large number of cis-edQTLs. Although the idea and approach are not novel (such as the study from Park et al. Genome Biology 2017, for A-to-I RNA editing and edQTL analysis across 445 human LCLs), this manuscript addresses an interesting topic in the field and utilizes a wide range of publicly available datasets. However, I have several concerns regarding the analysis approach and the data presented, which greatly dampens enthusiasm for the manuscript in its current form.

Major concerns:

1. It is well known that Cohen's d is frequently used in estimating sample sizes for statistical testing. Here, it is not clear why the authors applied Cohen's d when comparing AEI, the expression of ADAR genes, RNA hyper-editing sites, etc., among the MGE-GABA, GLU and OLIG groups, and what is the rationale by using Cohen's d when sample size is only 9 per group?

Thank you for this comment. The rationale of using Cohen's d (calculated by subtracting the means and dividing the result by the pooled standard deviation) is to emphasize two important observations: 1) that the effect sizes, as pertaining to RNA editing differences between the cell types, are significant and 2) these effect sizes were achievable across a moderate number of biological replicates. We further clarify this rationale in our revised Figure 1 legend.

2. I am surprised to see such a big difference in numbers of RNA hyper-editing sites and cell type-specific sites among MGE-GABA, GLU and OLIG. This is interesting. However, the authors didn't present basic reads and mapping stats for three groups of samples. Thus, it is impossible to assess whether sequencing yields and the quality of sequencing reads (e.g. mapping ratio and high quality mapping ratio) biased these numbers.

It is important to emphasize that RNA hyper-editing sites are normalized by the total number of mapped bases, as we and others have previously shown (see Porath et al., 2017, Genome Biology). This metric produces the number of hyper-edited bases per million mapped bases and is summarized as:

$$\text{Normalized hyper-editing signal} = (\text{Number of RNA hyper-editing sites} / \text{Total number of mapped bases}) * 1,000,000$$

This formula is elaborated upon in our revised Figure 1 legend and Materials and Methods section. Moreover, implementing Picard Tools, we now provide the total number of mapped bases for all samples in revised Supplemental Table 1. In this table, we also now summarize the total number of mapped bases split by ribosomes, protein-coding regions, UTR's, introns and intergenic regions, in addition to seven additional QC metrics. We tested the significance of these metrics between cell types and observed a higher number of bases mapping to intergenic regions for GLU cells. However, all other QC metrics were comparable across cell populations. See Figure and Table of *p*-values generated for the reviewer here:

P-values reported for pairwise comparisons						
	Total bases	Ribosomal bases	Coding bases	UTR bases	Intronic bases	Intergenic bases
OLIG vs. GLU	0.096	0.930	0.632	0.362	0.068	0.004
OLIG vs. GABA	0.714	0.229	0.248	0.483	0.789	0.347
GLU vs. GABA	0.104	0.230	0.496	0.212	0.086	0.037

Notably, sample H427 in the GLU group also shows a very low number of RNA hyper-editing sites compared to others in the same group. Is there any indication of the reason from this sample? The average number of RNA hyper-editing sites in the GLU group was apparently dragged down by this sample.

RIN, pH, Age and PMI are well-matched across samples from all donors, and these variables do not explain why H427_GLU sample displayed low hyper-editing rates. However, we do observe that the total number of mapped bases for this individual sample is lower relative to the others (H427_GLU = 6,008,890,255, remaining GLU samples ~18,791,035,534), implying lower coverage and read depth for this sample is justification enough for the lower RNA hyper-editing rate.

It is also worth noting that, in M&M (Line 144), the authors stated that sites should meet the criterion of in at least 8 out of 9 donors, which seems to compromise with this sample in the GLU group, but I am concerned that it may even cause more bias (of the difference) when comparing neuron groups with the OLIG group.

When finalizing GLU sites using the 8 out of 9 parameter, sample H427_GLU did not display excessive missingness (~15%, $n^{\text{sites}}=16,561$ out of 109,734 sites total). Next, for each missing site per sample, we computed the standard deviation of editing rates across the remaining samples and took the average standard deviation across all missing sites per sample – as an estimate of variance that might confound median

imputation for a group of sites in a specific sample. Importantly, the standard deviation of editing rates for these sites across the remaining 8 samples was very low (<9%) and was comparable with the average standard deviations of the missing sites for the remaining samples. See table summarizing these results:

Sample ID	# missing sites (% of GLU total)	Average standard deviation of editing rates across remaining samples with non-zero status
H406_GLU	6260 (5.7%)	Average SD of editing rates across these 6260 sites for the remaining 8 samples = 5.3%
H395_GLU	3048 (2.7%)	Average SD of editing rates across these 3048 sites for the remaining 8 samples =6.8%
H344_GLU	6315 (5.7%)	Average SD of editing rates across these 6315 sites for the remaining 8 samples =7.0%
H427_GLU	16561 (15.0%)	Average SD of editing rates across these 16561 sites for the remaining 8 samples =8.6%
H286_GLU	8747 (7.9%)	Average SD of editing rates across these 8747 sites for the remaining 8 samples =8.1%
H444_GLU	8215 (7.4%)	Average SD of editing rates across these 8215 sites for the remaining 8 samples =8.3%
H372_GLU	11269 (10.2%)	Average SD of editing rates across these 11269 sites for the remaining 8 samples =9.0%
H412_GLU	8588 (7.8%)	Average SD of editing rates across these 8588 sites for the remaining 8 samples =8.4%
H276_GLU	3715 (3.3%)	Average SD of editing rates across these 3715 sites for the remaining 8 samples =8.0%

Further, we repeated the differential RNA editing analysis between GLU and OLIG. However, this time dropping sample H427_GLU from the QC steps and differential comparison. We confirmed and reproduced our originally reported result in Figure 3 by measuring the concordance of the difference in editing rates between GLU and OLIG ($R^2=0.98$). See attached figure:

3. In Supplemental Table 1, the normalized HE counts seem very large. For example, the number of HE counts and normalized HE counts for sample H276 are 282599 and 16.99, respectively. Based on the definition of the “normalized HE counts” mentioned in the Results section (“we normalized the hyper-editing signal to the number of mapped bases per sample”), I can calculate the number of mapped bases for this sample is only 16,633 bp, which doesn’t make sense at all. The author should correctly describe how they generated the “normalized HE counts”. Again, the basic reads and mapping stats are essential to understand what was exactly happening here.

Thank you for pointing this out. We provide the detailed RNA-seq quality control metrics via Picard tools in our revised Supplemental Table 1. We also provide the equation used to compute the normalized hyper-editing signal (see the answer to Q2 above) in our revised text.

$$\text{Normalized hyper-editing signal} = (\text{Number of RNA hyper-editing sites} / \text{Total number of mapped bases}) * 1,000,000$$

4. Lines 130-131, the authors stated that “Hyper-editing sites commonly occurred in introns and 3’UTRs”.

Line 154: “Second, ~73% of editing events were detected in introns, while only a small fraction impacted protein coding regions (~0.67%) (Figure 2D)”.

It is a bit surprising to see very high signals in introns from RNA-Seq data. Are these all related to intron retentions? I assume most RNA-Seq reads should be aligned to the exon regions rather than the introns. However, there are no stats for me to check this information.

RNA editing sites are known to occur predominately in non-coding regions. In fact, among known sites catalogued in the REDiportal database ($n^{\text{sites}}=15,528,705$), ~58% of sites ($n^{\text{sites}}=9,117,562$) reside in introns whereas only ~.27% of sites ($n^{\text{sites}}=42,257$) reside in exons. Moreover, it is important to underscore that our RNA-seq data are from nuclei and, therefore, will inherently contain significantly more immature transcripts with retained intronic sequences than RNA-seq data generated from a cytoplasmic fraction or from bulk brain sample, as previously shown (Price et al., 2020, Genome Research; Zaghoor et al., 2021 Sci Reports). This has also been consistently shown across snRNA-seq and scRNA-seq studies using various tissues and model systems (Wu et al., 2019, JASN; Baken et al., 2018, PLOS One). Our results in revised Supplemental Table 1 are consistent with these reports, in that we find significantly more mapped bases in introns (~9,771,446,309/sample) relative to exons (~1,572,900,421/sample) ($p=9.2 \times 10^{-15}$). We discuss these results considering the enrichment of introns in our samples in our revised Discussion, specifically quoted here, “Moreover, intronic regions, which were highly edited in FANS-derived data, are often over-represented in RNA-seq and snRNA-seq generated from the nuclear fraction relative to a cytoplasmic fraction or bulk tissue sample, and has been demonstrated across various tissues and model systems³⁸⁻⁴⁰.”

In previous studies, such as Park et al (Genome Res. September 2012 22: 1626-1633) showed examples of overhanging RNA-Seq reads that were mapped incorrectly into the intron when the correct position should be in the adjacent exon, even though the splice junction was provided to the read mapper. It is not clear how many intron-derived sites were due to the above-mentioned misalignment.

One way to combat against this misalignment is to filter out sites that reside within 5bp of splice sites, as previously shown by Park *et al.*, 2012. We also applied this filtering step in our current work. For example, based on our *de novo* calling results, we do not detect any novel A-to-G sites within 5bp a splice site that survive our rigorous thresholding and filtering steps. Thus, we do not believe misalignment to be a considerable issue in the current study. We further elaborate on this step in our revised Materials and Methods, specifically quoted here, “RNA editing sites within 5bp of an annotated splice site were removed to avoid issues with mismapped reads that should have mapped across splice junctions”. In addition, it is worth noting that there has been substantial progress in updating genome builds, genome annotation files, RNA-seq mapping tools, and *de novo* calling algorithms since the Park *et al.*, 2012 report which may also explain a subset of these discrepancies in prior research.

In addition, it seems that the authors didn’t consider whether consecutive adenosines are more common in introns and 3’UTRs than other regions, which may also cause the bias of what the authors find in this study.

To address this comment, we collected 20,000 random RNA sequences per genic region, focusing on introns, intergenic regions, UTRs, and exons. Each RNA sequence was 1001bp in length and centered around a random RNA editing site reported in the REDiportal database, that also occurred in an *Alu* element (thus, 500bp were collected both upstream and downstream the editing event). Next, we investigated each set of sequences for the total percentage of adenosines out of all bases and found adenosines comprised ~26% of bases in UTRs, ~27% for exons, ~28% for intergenic regions, and ~29% for introns. While the differences in consecutive adenosines among these genic regions was subtle (<~3%), adenosines

were more common in introns compared to the other genic regions ($p < 2.0 \times 10^{-50}$ for all pairwise comparisons with introns, Wilcox test). However, the difference in consecutive adenosines between UTRs and exonic regions was insignificant ($p=0.58$, Wilcox test). Thus, while a very small fraction of total bases in introns are adenosines relative to other genic regions (~3%), we believe there are several additional biological factors that drive editing in introns in the current dataset.

We wish to highlight the following for further context: 1) In these experiments, both hyper-editing sites (from unmapped reads) and selective editing sites (from mapped reads) were more commonly located in introns followed by intergenic regions and UTRs, consistent with previous reports that RNA editing is more abundant in non-coding regions irrespective of whether a site is located along a hyper-edited transcript or not; 2) A leading reason for the intronic enrichment in the current study is that we are using nuclear RNA-seq data which contains a higher proportion of intronic reads (see the answer to Q4 above); 3) Existing reports using the hyper-editing approach in bulk tissues across various species and tissues also report an abundance of hyper-editing in similar non-coding regions (Porath et al., 2017, Genomic Biology); 4) hyper-editing requires a fairly complex double-strand RNA secondary structure for ADAR recognition and are also known to be enriched among hyper-edited regions (Porath et al., 2017, Genomic Biology; Porath et al., 2019, Nat Communications). Finally, to the Reviewer's point, we are using long-read isoform sequencing to systematically address these unique questions of RNA editing biology, including resolving hyper-editing clusters and double-stranded secondary structures – efforts outside the scope of the current manuscript.

5. The authors used publicly available snRNA-seq data to validate the RNA editing sites. However, the validation rates vary in three groups (from ~3% to ~27%). What does this variance mean? Does it indicate less accurate A-to-I call for the GABA and OLIG groups?

The variance in validation rates is largely a function of the total number of nuclei available to pool in the snRNA-seq data (Figure 4D). When more nuclei are subjected to snRNA-seq and pooled into a specific cell population for RNA editing quantification, we can observe higher validation rates with FANS RNA-seq. In our FANS RNA-seq data, we sequenced ~100,000 nuclei for each cell type and subject to an average of ~50M paired end reads per sample, relative to ~17,000 nuclei across three donors in our snRNA-seq data. Thus, we anticipate that as snRNA-seq datasets grow and we sequence more nuclei across a larger number of biological replicates, validation rates with techniques like FANS will be more consistently supported.

Moreover, I think it would be more informative if the authors can extract the number of sites that were not successfully validated but have enough coverage in the snRNA data for the estimation of false discovery rates.

Thank you for this great suggestion. Based on this comment, we made minor changes to define sites that validate as those with sufficient coverage and a median editing ratio within 50% of that reported by FANS RNA-seq (e.g. the absolute difference between the median editing ratio of a site between FANS and snRNA-seq is less than 50%). Subsequently, sites with sufficient coverage but a difference between median editing ratios that is greater than 50% were classified as sites with sufficient coverage, but without validation of RNA editing levels. These were then used to estimate the FDR. We now include these metrics

in our revised Supplemental Table 8, Materials and Methods and Results sections. For clarity, we also summarize these findings here:

	FANS		snRNA-seq		
Cell type	# sites	# nuclei	# sites with validation	# sites with coverage that do not validate	Estimated FDR
GLU	109,264	10,572	27,111 (24.8%)	1,456	0.05
GABA	107,999	2,075	11,509 (10.6%)	252	0.02
OLIG	64,374	1,709	2,996 (4.6%)	179	0.05

6. Lines 567-576: Residual (e.g. the resid function in the R package) is the difference between an observed value and the value predicted by a regression model. It is not clear why & how the authors adjusted the gene length using the resid function. To normalize the data by gene length, a more straightforward approach is to normalize the gene size to a certain length (e.g. 1 kb).

We modified this analysis in our revised Materials and Methods to report the density of RNA edits per gene normalized by gene length, as quoted here: “To adjust for gene length, we normalized the number of A-to-G editing events per gene for each cell type by the log of gene length (number of edits per gene/ $\log_2(\text{gene length}+1)$).” This produced results of cell-specific genes enriched with RNA editing sites that were highly similar to our previous analysis, now featured in our revised Supplemental Figure 10, as below:

Figure S10. RNA editing sites as a function of gene length. (A) Gene length as a function of the total

number of selective RNA editing sites across all MGE-GABA (left), GLU (center) and OLIG cells (right). **(B)** Pairwise comparisons of concordance for the number of RNA editing sites per gene between two cell types – MGE-GABA vs. GLU (left), MGE-GABA vs OLIG (center), GLU vs. OLIG (right). **(C)** Pairwise comparisons of concordance for the number of RNA editing sites by gene divided by log₂ gene length (+1) between two cell types – MGE-GABA vs. GLU (left), MGE-GABA vs OLIG (center), GLU vs. OLIG (right). Outlier genes were classified as genes enriched with RNA editing sites and are denoted as those outside the 99% confidence intervals from the grand mean.

Minor issues:

1. Some mistakes in writing:

i) “discovery rates consistent global editing activity (Figure 1)”

We have amended this to read: “*discovery rates consistent with higher global editing levels in neurons (Figure 1)*”.

ii) “Raw FASTQ files were obtained for a total of 27 paired-end (125 bp) nuclei samples were collected from”

We have amended this to read: “*Raw FASTQ files were obtained for 27 paired-end (125bp) nuclei...*”.

iii) “Raw FASTQ files were obtained for a total of 1431 paired-end (75 bp) samples were collected across 13 brain regions”

We have amended this to read: “*Raw FASTQ files were obtained for a total of 1431 paired-end (75 bp) samples across 13 brain regions...*”.

iv) Line 345: what is ‘200kb’ bp the target adenosine?

The majority of cis-edQTLs featured a SNP within 200kb (up or downstream) an editing site. We have modified our revised text accordingly.

v) the Figure legend of 7B: ‘+/-200bp’

We have amended this to read: “+/- 200kb”.

2. Lines 131-133: “and enriched for a local RNA editing sequence motif whereby guanosine is depleted -1bp upstream and enriched +1bp downstream the target adenosine (Figure S1D-E); validating the accuracy of our hyper-editing approach.”

I don’t see the point why the accuracy of our hyper-editing approach was validated here. If it is in line with previous reports, the authors need to correctly cite the relevant publications.

We apply computational approaches that we and others have implemented in prior-research. However, we believe that consistent ranking and annotation of sites to reveal hallmark features of editing will ensure robustness for this body of work and for the field moving forward.

We modified this sentence to read as: “...(Figure S1D-E); consistent with previous reports^{7,15}, thus further validating the accuracy of the hyper-editing approach.”

3. Line 146: “107,999” - should it be 107,998 based on Supplemental Table 2?

Thank you, we have modified this so it reads ‘107,998’.

4. What is the definition of “the 189,229 cell type-associated sites” mentioned in Abstract? It is not mentioned in M&M and Results sections.

These sites comprise the sum of all unique FANS-derived sites identified in the current study. This has been clarified in the revised Results section, specifically, “*Overall, we identified 189,229 cell type-associated RNA editing sites, including a total of 107,998 sites on 4,781 genes in MGE-GABA...*”.

5. Line 148: “We defined ~60% of these sites as cell type-specific as they were uniquely detected in one cell type”.

“~60% of these sites as cell type-specific” seems not correct. For example, in Figure 2A the venn diagram, GLU specific sites is obviously less than 40%.

Within each cellular population, the reviewer is correct that ~36% of MGE-GABA, ~36% of GLU and ~55% of OLIG sites classify as cell-type specific. Our estimate of ~60% was computed by using the sum of total number of cell-specific sites across MGE-GABA, GLU and OLIG cells (106,043 sites) over the total number of unique RNA editing sites identified across all cell types (189,229 sites). We have modified our text accordingly for clarity: “... *~36% of MGE-GABA, ~35% of GLU and ~55% of OLIG sites classify as “cell type-specific...”*”

6. In Fig 3B, for the cell type-enriched sites, the intersection of OLIG and GLU has two numbers, 11508 and 24. Please correct.

Figure 3 has been modified accordingly.

7. In GO enrichment analysis, it is not clear how the authors selected genes if the differentially edited sites were in intergenic or intron regions. Given these two categories contain most of the RNA editing sites, it will significantly affect the GO results.

Thank you for this suggestion. In testing this hypothesis, removal intergenic sites from the enrichment analysis had no significant effect on subsequent results, but removal of differentially edited intronic sites (~80% of all DE sites) had a substantial influence, removing many of the previously enriched terms. We now present these results in revised Supplemental Table 4 and provide justification for this recommended analysis in our revised Materials and Methods, specifically: “*Using this framework, we tested enrichment among i) all differentially edited sites and ii) all differentially edited sites with the exception of intronic sites (i.e. removing intronic sites, which were the majority) to gauge their overall influence on enrichment results.*”

8. Regarding the RNA recoding events, Supplemental Table 5 lists 84 sites, while the authors claimed there were 38 RNA recoding events (line 213). This is inconsistent. Please clarify.

We identified 84 known RNA recoding events, and 38 were significantly differentially edited between at least two cell types. This has been clarified in our Supplemental Table 5 file description.

9. Line 685: “To control for multiple tests, the FDR was estimated for all cis-edQTLs (defined as 1Mb between SNP marker and editing position)” So, what is the number of all cis-edQTLs for the multiple tests control?

We now provide these numbers for each brain region in revised Supplemental Table 11 (edQTL overview tab).

10. Line 688: what exactly does it mean by “brain promoter and enhancer regions”

We have modified our text to reflect that these regions indicate gene promoter and enhancer regions that are highly brain-specific, and have not been identified in other tissues. This collection of brain-specific gene promoters and enhancers has been generated through the FANTOM Slidebase Project (Ienasescu H et al., 2016, Database) and can be accessed here: <https://slidebase.binf.ku.dk/>

For additional clarity, we have modified this text in our revised Results: “*Notably, max-edQTL SNPs were moderately enriched in gene promoter and enhancer regions specific to the brain tested from the FANTOM project collected from SlideBase database (Supplemental Table 11).*”

Reviewer #2 (Remarks to the Author):

In this manuscript entitled “Cellular and genetic drivers of RNA editing variation in the human brain”, Cuddleston et al. studied cell type-specific RNA editing sites in human brain. They first called cell type-specific RNA editing sites in three major cell populations including medial ganglionic eminence (MGE) - derived inhibitory GABAergic interneurons (MGE-GABA), excitatory glutamatergic neurons (GLU), and oligodendrocytes (OLIG) from adult prefrontal cortex (PFC) by fluorescence-activated nuclei sorting (FANS). These results showed that neuron-specific editing sites are of higher editing levels than those of OLIG, and cell-specific editing sites are related to different biological pathways. Then they validated these results by single-nuclei RNA sequencing. Next they further verified that RNA editing levels are correlated with neuron proportions from 13 brain regions by using data from GTEx project. Finally, they identified genetic variants associated with RNA editing levels. This work provides lines of evidence that RNA editing levels are correlated with cell types in brain, indicating the potential biological regulation roles of RNA editing in different brain cell types. The manuscript is well organized and written. There are only several minor suggestions for the authors to consider during revision.

1. The authors analyzed RNA editing sites among three major cell populations (MGE-GABA, GLU, and OLIG) purified from adult prefrontal cortex (PFC). It will be much appreciated if the authors could specify why not choose other than PFC regions for the analysis. In addition, since at least 24 unique cell populations were detected in PFC (Figure 4A), the authors may also want to explain a little bit how many of them belong to MGE-GABA or GLU, respectively, or their proportions in PFC.

Our study offers a fine-grained perspective on A-to-I editing within the PFC, the brain region that is critical for cognition, memory, and executive function and is broadly implicated in neuropsychiatric illness. We provide further justification of studying the PFC in our revised Introduction. Specifically, we now state:

“We first quantified RNA editing among three major cell populations purified from the adult prefrontal cortex (PFC)--the brain region that is critical for cognition, memory, and executive function and is broadly implicated in neuropsychiatric illness (PMID: 17726913). The PFC contains two major types of neuronal populations, the excitatory glutamatergic (GLU) and the inhibitory GABAergic interneurons, which account for about 80% and 20% of all cortical neurons, respectively (PMID: 24097041). Medial ganglionic eminence (MGE)--derived interneurons comprise ~60 to 70% of all cortical GABAergic neurons and contain parvalbumin- and somatostatin-expressing interneurons that have been implicated in neurodevelopmental and neuropsychiatric disorders (PMID: 23395369, 24429630). Oligodendrocytes (OLIG) are the major glial cell type in the central nervous system that provides support and myelin-based insulation to axons (PMID: 11274346).”

In terms of intersecting snRNA-seq nuclei with FANS cell populations, in our revised Results and Figure 4 legend, we provide exact numbers in terms of how many individual nuclei and which cell populations from snRNA-seq belong to the MGE-GABA, GLU and OLIG populations. In brief: *“We observed that ~52% of all nuclei expressed the markers of our GLU population (Rbfox3+/Sox6- $n^{\text{nuclei}}=8957$), ~21% expressed the MGE-GABA marker (Rbfox3+/Sox6+ $n^{\text{nuclei}}=3708$), and ~9% expressed the OLIG markers (Rbfox3-/Sox10+, $n^{\text{nuclei}}=1709$). Note that our FANS GLU population includes mainly glutamatergic neurons, but also contains a small proportion of CGE-derived GABAergic neurons.”*

2. In Line 180-185, about 400 differential expressed RBPs were shown to be correlated with global editing activity, however only FMRP was chosen for further eCLIP validations. Why?

Thank you for your question. While several RBPs were determined to be differentially expressed between our FANS-derived cell populations, FMRP is currently the only RBP with publicly available eCLIP-seq data for validation of overlapping RBP hotspots with cell-specific RNA editing sites. Importantly, FMRP is also known to interact with ADAR (Shamay-Ramot et al., 2015, PLoS Genetics). For context, the ENCODE project has published eCLIP-seq data for ~150 RBPs. However, the assay was performed on K562 and HepG2 cells, and shows substantial cell type-specific differences (Quinones-Valdez et al., 2019, Communications biology). For these reasons, we did not include this data in our study. Further, the POSTAR database has a collection of CLIP-seq data but there is currently a dearth of experiments applied to human postmortem brain tissue. We further elaborate on this reason in our revised Materials and Methods section, quoted here, *“At the time of writing this paper, this experiment represents the only eCLIP-seq data generated from human postmortem brain tissue.”*

3. In Line 186-190 and Figure S4D-E, the authors got the conclusion that FMRP eCLIP peaks were significantly enriched for MGE-GABA (permutation p-value=0.005, Z-score=9.3) and GLU sites (permutation p-value=0.005, Z-score=6.2), and moderately enriched for OLIG sites (permutation p-value=0.005, Z-score=7.4). As the Z-score of GLU sites is smaller than that of OLIG sites, it is not clear how the authors concluded that GLU sites are more enriched than OLIG sites?

FMRP eCLIP-seq peaks were available across two technical replicates, however enrichment results for only one of the replicates was quoted in the text, while the results for both replicates were provided in the Supplemental Figure. We greatly clarify this result in our revised Results section, quoted here: *“We further*

examined whether FMRP binding sites were enriched near MGE-GABA, GLU and OLIG editing sites using enhanced ultraviolet crosslinking and immunoprecipitation (eCLIP) across two technical replicates in human frontal cortex. FMRP eCLIP peaks were significantly enriched for MGE-GABA (Rep₁, Z-score=9.3; Rep₂, Z-score=4.7), GLU (Rep₁, Z-score=6.2; Rep₂, Z-score=3.7), and OLIG sites (Rep₁, Z-score=7.4; Rep₂, Z-score=2.8) (Figure S5D-E)."

4. In Line 249-250, authors mentioned that "genes with higher rates of editing in MGE-GABA and GLU cells were not associated with cell-specific differential gene expression". If the editing didn't influence gene expression, what is the biological functions of editing? Whether splicing or secondary structures were regulated by these neuron-specific editing sites?

While RNA editing that directly regulates gene expression might not serve as the 'general rule' in these data, we do document hundreds of genes with concordant patterns of RNA editing and gene expression. We have clarified this in our revised Results section, quoted here: "*Cell type-enrichment differences in RNA editing rates explained ~7%, ~7%, and ~2% cell-specific differential gene expression in MGE-GABA ($n^{\text{genes}}=775$), GLU ($n^{\text{genes}}=660$) and OLIG ($n^{\text{genes}}=378$) populations, respectively (Figure S7).*"

Furthermore, and to address the reviewers point, we now provide an additional analysis that seeks to elucidate the probability of an RNA editing site to be splice altering. We applied SpliceAI, a deep neural network that predicts splice junctions from an arbitrary pre-mRNA transcript sequence. This approach enables precise prediction of non-coding RNA editing events that cause cryptic splicing and generates a delta score for each RNA editing site. The delta score is defined as the probability of the site being splice-altering and ranges from 0 to 1. In the primary publication of this work, a detailed characterization is provided for 0.2 (high recall), 0.5 (recommended), and 0.8 (high precision) cutoffs. In applying this approach, we identified a small subset of cell type-associated RNA editing sites that are characterized as splice altering, as quoted here: "*Moreover, we identified a small subset of cell type-enriched RNA editing sites predicted to be splice altering in MGE-GABA ($n^{\text{sites}}=28$), GLU ($n^{\text{sites}}=23$) and OLIG ($n^{\text{sites}}=23$) populations (Supplemental Table 4).*" We provide these results in revised Supplemental Table 4 along with corresponding new Material and Methods. We expect this number to be a low estimate of splice altering RNA editing sites because our quality control steps removed RNA editing sites within 5bp of known splice (see R1 comments). These sites were removed to avoid overhanging RNA-seq reads that were mapped incorrectly into the intron when the correct position should be in the adjacent exon (as described by Park *et al.*, 2012, Genome Research).

5. In Figure 4C, the ADAR1 expression level of endothelial was higher than those in five types of cells including two type of neurons (MGE-GABA and GLU), but the editing levels in endothelial were not comparable to those in two types of neurons (MGE-GABA and GLU). Any explanation?

In peripheral tissues and cells, A-to-I editing is largely ADAR1-dependent (Lamers *et al.*, 2019, Front. Immunol.; Pujantell *et al.*, 2017, Sci. Rep). In the CNS, A-to-I editing is largely ADAR2-dependent and, to a lesser extent, ADAR1-dependent (Horsch *et al.*, 2011, J Biol Chem.; Tan, 2017, Nature; Behm *et al.*, 2017, J Cell Sci). In our snRNA-seq experiment, both ADAR2 expression and the AEI are elevated in neurons relative to all other cell types. Thus, if RNA editing is catalyzed in endothelial cells as it is in most CNS cell types, in that is largely ADAR2-dependent, this may explain elevated expression of ADAR1

despite the lower AEI. Yet, given that endothelial cells in the CNS are affected by inflammation (e.g., PMID 29973684), is intriguing to speculate on whether RNA editing in these cells might be more ADAR1-dependent as observed in other peripheral and inflammatory cells – but independent experimentation would be required to test these separate hypotheses. We also want to emphasize that endothelial cells make up a very small number of total nuclei in the current study ($n^{nuclei}=464$, ~2.6% of all single nuclei), and future studies should ideally include a larger number of endothelial cells to better support these hypotheses.

6. Generally, snRNA-sequencing uses a 3'-end capture strategy for RNA enrichment and sequencing, and thus only parts of genes/3'UTR (about 200~500 bp long) could be sequenced. If it were the case, RNA editing calling from snRNA-sequencing might be incomplete/skewed due to the 3' bias. The authors at least need to discuss this point in the manuscript.

Thank you for this important critique. We have taken steps to quantify whether RNA editing sites that validate based on snRNA-seq data are skewed toward the 3' or 5' end of a gene. In brief, for each gene we collect the entire gene length and then compute the “5' end” as the first quartile of the total gene length and the “3' end” as the final quartile of the total gene length. Subsequently, using these bins, we asked whether RNA editing sites that validate in snRNA-seq reside towards the 5' or 3' end of a transcript. We tested this overlap using the regioneR R package that will compute a permutation-based significance of an overlap based on two sets of genomic coordinates. We used 200 random permutations and report the resulting number of overlapping sites, p-value and z-scores for GABA, GLU and OLIG validating sites that reside towards the 3' or 5' of a transcript. We found equal enrichment of validated RNA editing sites by snRNA-seq that reside on both 3' and 5' ends. These findings are now included in our revised Results section and quoted here: “*Sites with validation by snRNA-seq were equally enriched on 3' and 5' ends of the corresponding transcript for MGE-GABA, GLU and OLIG populations (Supplemental Table 8).*”.

	GABA			GLU			OLIG		
	Number of editing sites	P-value	Z-score	Number of editing sites	P-value	Z-score	Number of editing sites	P-value	Z-score
5' end	2996	0.009	8.2	6899	0.009	10.6	795	0.005	4.7
3' end	2966	0.005	7.4	6477	0.005	8.4	939	0.005	5.5

Some minor changes:

In Figure 3B, the overlap between OLIG and GLU wrongly occurred twice. The “Alu” should be italic, and please check throughout the manuscript.

Thank you, we have amended Figure 3 and italicized *Alu* throughout our revised manuscript.

In Figure 1C, 5C, 7G, the correlation coefficients (R2) are somehow mistakenly labelled less than 0 (for example, R2=-0.33 in Figure 1C).

Thank you, this has also been modified throughout our revised manuscript.

In Figure 4A, what are the full names of ExN and InN cells.

We have modified this in revised Figure 4 legend. In brief, excitatory (ExN) and inhibitory (InN) cell populations.

In Line 180-185, the name of RBPs such as “NOP14, PTEN and DYNC1H1” shouldn’t be italic.

Thank you, this has been modified.

In Line 345, “200kb bp the target adenosine” missed the range number. In addition, is 200kb too far away as cis?

This has been changed to read as 200kb. In this work, we set the edQTL search space as 1Mb within an editing site, similar to the majority of large-scale eQTL investigation. Here, we show that most edQTLs actually reside much closer than 1Mb, and the closer the lead SNP is to the editing site, the more significant the edQTL association, as we have shown previously (Breen et al., 2019, Nat Neuroscience).

Reviewer #3 (Remarks to the Author):

The manuscript by Breen et al. has used the FANS and single-nuclei RNA-sequencing ways to study the RND editing sites across the glutamatergic, MGE GABAergic neurons, and the oligodendrocytes cells from the human prefrontal cortex. The authors found 189,229 cell type-associated RNA editing sites and the neurons have a higher number of editing sites than oligodendrocytes. In addition, 661,791 cis-editing quantitative trait loci from various brain regions were discovered. This study provided the human RNA editing sites atlas and showed the cell type-specific editing sites, however, except some novel editing sites were found, no major outstanding discoveries were found in this study. I have some comments that I hope will help further improve this paper.

We sincerely thank the reviewer for the constructive comments. We believe that mapping cell-specific editing in the major brain cell types, as well as linking cell types to editing variation and genetic regulatory effects across multiple brain regions, represent a substantial advancement in the field. We have taken strides to clarify the importance of this research by emphasizing substantial existing knowledge gaps in the field within our revised Introduction.

Major comments:

1. The manuscript mainly presented the discovered RNA editing sites and the sites number comparison between the three types of cell types, however, the findings in this work are not new since the vast majority of RNA editing sites have been detected in the published works, including the finding of editing levels in neurons are higher than glial cells. From my point of view, if the authors can highlight some of the novel found RND editing sites, and explain the internal mechanism of why the sites are higher in neurons relative to oligodendrocytes, and linking the newfound sites to the regulation of neuronal transcription, splicing, and functionality.

Broadly speaking, we see the value of the current body of work as: 1) This is the first study to reveal >180,000 cell-type associated RNA editing sites in the brain. Prior to this work, there had been no cellular

resolution or general understanding of how discrete cell types might regulate RNA editing levels in the human brain; 2) We identified >600K edQTLs and hundreds with cell-type associated features. This is ~13-fold more edQTLs than previously documented, the majority in 3'UTRs, and hundreds that are unique to a specific cell type; 3) These findings do not just advance basic science regarding context-dependent regulation of RNA editing in the human brain, but also enable more precise translational studies (e.g. differential RNA editing patterns, explaining GWAS risk loci through effects on edQTLs *etc.*). A prerequisite to a more thorough investigation of any such mechanism is the precise identification of cell type- and genetically-regulated RNA editing sites. The data set we created addresses this need.

More specifically, we have clarified some of the important functional effects of RNA editing in the current work: 1) First, we clarify the influence of RNA editing on neuronal transcription and gene expression in our revised Results, where a small subset of gene expression profiles is explained by cell-specific RNA editing. These results are quoted here: “*Cell type-enrichment differences in RNA editing rates explained ~7%, ~7%, and ~2% cell-specific differential gene expression in MGE-GABA ($n^{\text{genes}}=775$), GLU ($n^{\text{genes}}=660$) and OLIG ($n^{\text{genes}}=378$) populations, respectively (Figure S7).*”; 2) Second, we applied a new statistical method called SpliceAI, a deep neural network that predicts splice junctions from an arbitrary pre-mRNA transcript sequence, thus computing a probability of an RNA editing site to be splice altering. We found a small subset of RNA editing sites predicted to be splice altering. We have added these findings to our revised Results section, quoted here: “*Moreover, a small subset of cell type-enriched RNA editing sites were predicted to be splice altering in MGE-GABA ($n^{\text{sites}}=28$), GLU ($n^{\text{sites}}=23$) and OLIG ($n^{\text{sites}}=23$) populations (Supplemental Table 4).*” We also detail these approaches in our revised Materials and Methods, quoted here, “*We applied SpliceAI to cell type-associated intronic sites in Supplemental Table 2. SpliceAI is a deep neural network that accurately predicts splice junctions from an arbitrary pre-mRNA transcript sequence. We used a delta score probability of 0.5 or higher as a threshold for an RNA editing site being splice-altering. Notably, we removed sites within 5bp of a known splice site before running this approach, and thus results likely represent a low estimate of truly splice altering events.*” 3) Finally, regarding additional biologically plausible explanations as to why neurons exhibit a high amount of RNA editing relative to glial cells, we provide a detailed account of this, as well as some plausible hypotheses in our Discussion, quoted here: “*Finally, while our data provide new avenues for understanding cellular specificity RNA editing, one key question remains: ‘What is the biological explanation as to why neurons exhibit a preponderance of RNA editing activity?’. Here we extend some putative solutions. One possible explanation of enhanced RNA sequence diversity and ADAR expression in neurons might be to afford increased neural plasticity. Neurons, more than other cell types, must be able to quickly respond to altered environmental inputs, therefore, RNA editing may represent a mechanism capable of aligning with the timing required for experience-dependent plasticity, similar to other RNA modifications^{39,40}. As such, editing activity in neurons might play a role in controlling the organizational architecture of neuronal networks implicated in higher order learning and memory. A related explanation of enhanced diversity of RNA editing in neurons may hint at its potential relation with essential neuronal functions. This idea is reinforced by the high density of editing in neuronal transcripts that encode proteins directly involved in excitatory/inhibitory (E/I) functions and the significant differences in editing between excitatory GLU and inhibitory GABA neurons. Such differences may modulate E/I balance within the cortical circuitry. Imbalance of E/I activity is thought to play a critical role in the pathophysiology of several different neuropsychiatric and neurological disorders, including autism spectrum disorder, schizophrenia, and epilepsy^{41,42}. Thus, functional relevance of the observed RNA editing differences between different*

populations of brain cells in health and disease warrants further investigation. Lastly, it remains unclear how RNA editing may play out across distinct cellular compartments and locations. Recent work has found discrete gene expression differences across synapses, dendrites, axons and neuronal bodies^{43,44}, and RNA editing may represent a driver of activity-induced RNA localization^{1,2,11}.”

2. Could you explain why the expression of ADAR1 in GLU cells is significantly higher than MGE-GABA neurons, while the opposite results are found for the ADAR2 between these two cell types in the Figure 1B? which seems are not matchable to Figure 1C. And why the oligodendrocyte cells have the highest expression profiles of ADAR3 than neurons in Figure 1B?

To clarify, ADAR1 is not more highly expressed in GLU relative to MGE-GABA ($p=0.08$) – the expression level is comparable between these two neuronal classes. However, we do see higher expression of ADAR2 in MGE-GABA relative to GLU populations ($p=0.0002$). This result is on par with our observations of higher global selective editing rates and hyper-editing rates in MGE-GABA cells relative to GLU and OLIG populations.

For further context, ADAR1 and ADAR2 have demonstrated catalytic activity and participate in A-to-I editing; in contrast, no editing activity has been detected with ADAR3 on known substrates and it appears to be catalytically inactive (Chen et al., 2000, RNA; Oakes et al., 2017, J Biol Chem). ADAR2 is key to A-to-I editing in the CNS and critical for neurological function, whereas ADAR1-mediated editing has an essential role in peripheral systems and in the prevention of activation of the cytosolic dsRNA innate immune sensing system by endogenous RNA (also see Q5 and response from Reviewer 2). For example, mice lacking *Adar2* die within 3 weeks of birth with increasingly severe seizures associated with effects of increased influx of calcium ions through unedited AMPA glutamate receptors on synaptic plasticity across the brain (Higuchi, M. et al, 2000, Nature; Krestel, H. E. et al., 2004, J. Neuro). Moreover, ADAR2 (as well as ADAR3) is most highly expressed in the brain and CNS and is restricted in its expression in other tissues. Thus, we anticipate that much of the RNA editing variation observed across cell types in the current study is largely ADAR2-dependent. This is also consistent with our recent large-scale meta-analysis of RNA editing across brain development and model systems, where ADAR2 explains most editing changes throughout development (Cuddleston et al., 2021, BioRxiv, under review). Regarding the expression of ADAR3 in OLIG populations, we hypothesize that the low editing rates in OLIG may be explained by the high expression ADAR3 outcompeting ADAR2 for dsRNA binding space, thus driving editing levels downward. It is intriguing to speculate on potential functions of ADAR3 in the brain, but future experimental follow-ups are required to delineate specific functional effects.

3. In Figure 2H, the authors showed that the found novel A-to-G sites displayed significantly more supporting read coverage compared to known sites, could you give the possible explanations? Did the authors found the same significance for the novel found C-to-T and G-to-A editing sites?

Thank you for this suggestion. We now introduce new Supplemental Figure 3 (below) to illustrate that RNA editing sites are commonly detected on low-to-moderately expressed genes. We also observe a larger percentage of novel sites being detected on lowly expressed genes, suggesting that these sites have been missed or subsumed by bulk brain RNA-seq sampling techniques.

Figure S3. RNA editing sites are commonly detected on low-to-moderately expressed genes. (A) The total number of genes with at least one RNA editing site (y-axis) relative to the corresponding expression level (\log_2 CPM) of the gene (x-axis). (B) The percentage of genes with at least one RNA editing site (y-axis) according to binned gene expression (x-axis) parsed by novel sites and known sites detected in the REDiportal database.

Further, we are unable to compare read coverage between known and novel C-to-T or G-to-A sites, because RNA editing databases (like REDiportal) only catalogue A-to-G events; thus all sites other than A-to-G in the current study will be classified as ‘novel’ by default. Nevertheless, we did compare read coverage across novel A-to-G, C-to-T and G-to-A sites, and found a comparable level of read coverage across these events. See Figure generated for the reviewer, here:

4. In Figure3A, the authors performed the differential editing analysis of the cell type-enrichment of RNA editing levels, some differentiated sites were marked on the volcano plots, but I didn’t find some text explanation of these differentiated sites, could the authors explain why these sites were marked, did these sites showing some linking to the cell type-specific molecular functionality?

We have removed these marked genes/sites from the volcano plots to avoid any confusion. They reflected genes harboring some of the most significant changes in our analysis.

5. The details of the preprocessing, normalization, dimension reduction, and clustering for the single nuclei sequencing data analysis should be included in the method part.

Thank you, we have added these details to our revised Materials and Methods, quoted here, “*In brief, a cell was excluded if the number of expressed genes less than 300 or more than 7000, or with the number of UMI less than 300 or more than 20000, or the percentage of mitochondria reads more than 5%. The normalization method was “LogNormalize” and the scale factor was 10000. The linear regression was performed by choosing the percentage of mitochondria reads as variable. Rfrom was used to compute the specificity score for each gene in each cell cluster. Hierarchical clustering was manually checked*

along with the top ranked genes in each cell cluster to determine cellular specificity based on well-known gene markers to verify the assignment of cell types and subtypes. Cells with inconsistent or no assignment were removed from the analysis.”

Minor comments:

1. is the full name of ADAR (adenosine acting on RNA) correct? Should it be Adenosine deaminases acting on RNA?

Thank you, this has been modified to - adenosine deaminase acting on RNA (ADAR).

2. statistical analysis details such as method and p-value threshold selection need to be given in the method part.

Thank you, we provide detailed methods and *p*-value thresholds throughout our Materials and Methods section and revised figure legends.

3. could you show the statistical analysis values in Figure 2I?

Yes, we now provide those directly in Figure 2I and include methods in the corresponding legend.

REVIEWER COMMENTS

Reviewer #1 (Remarks to the Author):

“2. I am surprised to see such a big difference in numbers of RNA hyper-editing sites and cell type-specific sites among MGE-GABA, GLU and OLIG. This is interesting. However, the authors didn't present basic reads and mapping stats for three groups of samples. Thus, it is impossible to assess whether sequencing yields and the quality of sequencing reads (e.g. mapping ratio and high quality mapping ratio) biased these numbers.

It is important to emphasize that RNA hyper-editing sites are normalized by the total number of mapped bases, as we and others have previously shown (see Porath et al., 2017, Genome Biology). This metric produces the number of hyper-edited bases per million mapped bases and is summarized as: This formula is elaborated upon in our revised Figure 1 legend and Materials and Methods section. Moreover, implementing Picard Tools, we now provide the total number of mapped bases for all samples in revised Supplemental Table 1. In this table, we also now summarize the total number of mapped bases split by ribosomes, protein-coding regions, UTR's, introns and intergenic regions, in addition to seven additional QC metrics. We tested the significance of these metrics between cell types and observed a higher number of bases mapping to intergenic regions for GLU cells. However, all other QC metrics were comparable across cell populations. See Figure and Table of p-values generated for the reviewer here...”

Follow-up comments:

Regarding the numbers of RNA hyper-editing sites and cell type-specific sites among MGE-GABA, GLU and OLIG, I was referring to Figure 1D.

Unfortunately, providing the total number of mapped bases for all samples doesn't really help to assess the quality of sequencing data. For example, one may receive similar number of mapped bases for the below two scenarios:

- i) Sample A has 100 million sequencing reads, but it has a rather poor sequencing quality (could be due to the low quality of libraries or any failed steps during the sequencing procedure), which ends up ~30% of mapping ratio and even lower ratio if mapping quality (e.g. above Q20) is considered.
- ii) Sample B has 30 million sequencing reads, and it has very good quality of library prep & sequencing, with almost all reads could be successfully aligned to the genome.

However, “similar number of mapped bases” here may have large biases to the detection of RNA hyper-editing sites due to the fact that sequencing yields and/or their qualities are very different.

“Notably, sample H427 in the GLU group also shows a very low number of RNA hyper-editing sites compared to others in the same group. Is there any indication of the reason from this sample? The average number of RNA hyper-editing sites in the GLU group was apparently dragged down by this

sample.

RIN, pH, Age and PMI are well-matched across samples from all donors, and these variables do not explain why H427_GLU sample displayed low hyper-editing rates. However, we do observe that the total number of mapped bases for this individual sample is lower relative to the others (H427_GLU = 6,008,890,255, remaining GLU samples ~18,791,035,534), implying lower coverage and read depth for this sample is justification enough for the lower RNA hyper-editing rate."

Follow-up comments:

If this were the case, then why wouldn't "H427_GABA" and "H395_GABA" show the similar trend as we found in the "H427_GLU" sample? Notably, the total number of mapped bases of "H427_GABA" and "H395_GABA" are 2,257,772,021 and 6,274,393,384, respectively, remaining GABA samples ~14,608,389,571.

Reviewer #2 (Remarks to the Author):

In the revised manuscript entitled "Cellular and genetic drivers of RNA editing variation in the human brain", the authors have addressed most of my concerns, but not the concern #6.

Since snRNA-seq uses a 3'-end capture strategy, it can be misleading to state that "numbers of validated RNA editing sites in 5' and 3' ends are equal". The authors can first align reads across gene bodies with both snRNA-seq or bulk-seq datasets, and then they will see a 3'-bias in snRNA-seq, but not in bulk-seq. In this case, how could the RNA editing sites in 5' end be precisely validated with a 3'-biased snRNA-seq? Please further address this remaining concern.

Reviewer #3 (Remarks to the Author):

I have no more questions.

We sincerely thank the reviewers for their generous time and careful review of our work. We provide point-by-point responses to all minor critiques below.

Reviewer #1 (Remarks to the Author):

Previous comment: “I am surprised to see such a big difference in numbers of RNA hyper-editing sites and cell type-specific sites among MGE-GABA, GLU and OLIG. This is interesting. However, the authors didn’t present basic reads and mapping stats for three groups of samples. Thus, it is impossible to assess whether sequencing yields and the quality of sequencing reads (e.g. mapping ratio and high quality mapping ratio) biased these numbers.

Previous response: It is important to emphasize that RNA hyper-editing sites are normalized by the total number of mapped bases, as we and others have previously shown (see Porath et al., 2017, Genome Biology). This metric produces the number of hyper-edited bases per million mapped bases and is summarized as: This formula is elaborated upon in our revised Figure 1 legend and Materials and Methods section. Moreover, implementing Picard Tools, we now provide the total number of mapped bases for all samples in revised Supplemental Table 1. In this table, we also now summarize the total number of mapped bases split by ribosomes, protein-coding regions, UTR’s, introns and intergenic regions, in addition to seven additional QC metrics. We tested the significance of these metrics between cell types and observed a higher number of bases mapping to intergenic regions for GLU cells. However, all other QC metrics were comparable across cell populations. See Figure and Table of p-values generated for the reviewer here...”

R1 follow-up comments:

Regarding the numbers of RNA hyper-editing sites and cell type-specific sites among MGE-GABA, GLU and OLIG, I was referring to Figure 1D. Unfortunately, providing the total number of mapped bases for all samples doesn’t really help to assess the quality of sequencing data. For example, one may receive similar number of mapped bases for the below two scenarios: i) Sample A has 100 million sequencing reads, but it has a rather poor sequencing quality (could be due to the low quality of libraries or any failed steps during the sequencing procedure), which ends up ~30% of mapping ratio and even lower ratio if mapping quality (e.g. above Q20) is considered. ii) Sample B has 30 million sequencing reads, and it has very good quality of library prep & sequencing, with almost all reads could be successfully aligned to the genome.

However, “similar number of mapped bases” here may have large biases to the detection of RNA hyper-editing sites due to the fact that sequencing yields and/or their qualities are very different.

Thank you kindly for the clarification. We now include additional RNA-seq and alignment QC metrics for each sample (generated using Picard tools) in revised **Supplemental Table 1**. To address reviewer's concerns about the sequencing yields and quality of sequencing reads, here we show that the fraction of passed filtering (PF) high quality (HQ) aligned bases is comparable across all cell types in the current study. **Supplemental Table 1** contains supporting details and further indicates that the total number of reads, PF reads, PF aligned reads, PF HQ aligned reads with $Q > 20$, along with several additional QC metrics, were also comparable across cell populations, with no significant differences.

Previous comment: "Notably, sample H427 in the GLU group also shows a very low number of RNA hyper-editing sites compared to others in the same group. Is there any indication of the reason from this sample? The average number of RNA hyper-editing sites in the GLU group was apparently dragged down by this sample."

Previous response: RIN, pH, Age and PMI are well-matched across samples from all donors, and these variables do not explain why the H427_GLU sample displayed low hyper-editing rates. However, we do observe that the total number of mapped bases for this individual sample is lower relative to the others (H427_GLU = 6,008,890,255, remaining GLU samples ~18,791,035,534), implying lower coverage and read depth for this sample is justification enough for the lower RNA hyper-editing rate."

Follow-up comments:

If this were the case, then why wouldn't "H427_GABA" and "H395_GABA" show the similar trend as we found in the "H427_GLU" sample? Notably, the total number of mapped bases of "H427_GABA" and "H395_GABA" are 2,257,772,021 and 6,274,393,384, respectively, remaining GABA samples ~14,608,389,571.

The reviewer is correct that we do observe a similar trend for this subset of samples, and this is now accurately reflected in revised **Supplemental Table 1** and **Figure 1** (as noted above). As for explaining the remaining amount of RNA editing site detection/variation per donor, several factors account for a large fraction of this variation (certainly not all of it), including variation in ADAR1 and ADAR2 expression (see **Figure S1**) and/or common genetic variation (as in **Figure 7**). Substantial inter-donor variability in RNA site detection is common for large-scale RNA editing studies in the CNS, and clarifying the remaining RNA editing variance after accounting for enzymatic, cellular, and genetic differences is a key future step for our research program.

Reviewer #2 (Remarks to the Author):

In the revised manuscript entitled "Cellular and genetic drivers of RNA editing variation in the human brain", the authors have addressed most of my concerns, but not the concern #6.

Since snRNA-seq uses a 3'-end capture strategy, it can be misleading to state that “numbers of validated RNA editing sites in 5' and 3' ends are equal”. The authors can first align reads across gene bodies with both snRNA-seq or bulk-seq datasets, and then they will see a 3'-bias in snRNA-seq, but not in bulk-seq. In this case, how could the RNA editing sites in 5' end be precisely validated with a 3'-biased snRNA-seq? Please further address this remaining concern.

The reviewer is correct that our snRNA-seq data are 3' biased. To confirm, we now include the RNA-seq and alignment QC metrics for all snRNA-seq pools in revised **Supplemental Table 1**, confirming a high 3' bias of mapped snRNA-seq reads and editing sites. We also generated new **Supplemental Figure 12** to support the interpretation of these results, which are summarized in our Results section and quoted here: “Sites with validation by snRNA-seq were biased towards the 3' end of the transcript, and those with higher supporting read coverage displayed a stronger 3' bias relative to those with lower coverage (**Figure S12**)”. This figure captures the following:

1. We now clearly illustrate differences in read coverage distributions from 5' to 3' transcript ends between FANS RNA-seq and snRNA-seq (**Supplemental Figure 12A**).
2. To streamline interpretation, we re-computed our 3' bias by measuring distance between each editing site and the transcription start site (TSS) of the respective gene. Using this simple distance metric, we parsed all RNA editing sites that validated by snRNA-seq into three bins: sites with high, medium and low supporting snRNA-seq read coverage. Differences in distance to the TSS were evaluated using a Mann Whitney-U test for sites in each bin relative to sites without snRNA-seq validation (**Supplemental Figure 12B**). Sites that validate by snRNA-seq with highest supporting read coverage show the strongest 3' bias.
3. These methods are also described in our revised Materials and Methods section.

Figure S12. Read bias distribution of FANS and scRNA-seq. (A) RseqQC computed RNA-seq read coverage over the entire gene body for all transcripts for the FANS RNA-seq data (right) and the snRNA-seq pools (left). The average coverage rates across all replicates per cell type are plotted. **(B)** Sites that validate by snRNA-seq were evaluated for

3' bias and were binned into three groups based on total snRNA-seq read coverage: 1) high coverage (the first quintile of coverage); 2) moderate coverage (the second, third and fourth quintile of coverage); and 3) low coverage (the fifth quintile of coverage) (y-axes). The distance between each site and the transcription start site (TSS) was measured (x-axis) for MGE-GABA (left), GLU (center) and OLIG (right). Mann Whitney-U tests were used to examine significant differences in distances from the TSS for all binned sites that validate by snRNA-seq relative to sites without validation, whereby sites with higher read coverage show a greater 3' bias.

Reviewer #3 (Remarks to the Author):

I have no more questions.

REVIEWER COMMENTS

Reviewer #1 (Remarks to the Author):

There are more concerns in this revision.

1) The authors have now provided the total number of reads and the alignment stats. However, I am surprised to see that the alignment ratio is always 100% for every sample, i.e. "PF_READS" equals to "PF_READS_ALIGNED" in Supplemental Table 1 – "RNAseq QC and alignment metrics" sheet, which is too good to be true.

2) I am very concerned that the authors revised some fundamental results/data in Supplemental Table 1 without any explanations. Furthermore, these revised results should lead to different downstream outcomes and/or conclusions. However, the authors only made changes in Supplemental Table 1, but not any other results in their study.

For example, the number of total hyper editing sites (the "Total.hyperEditing.Sites" column) was revised from 331,780 to 31,779 for sample GABA-H395. That's TEN times drop! In addition, this number was revised from 456,934 to 72,933 for sample GABA-H427.

Please see the attachment for the screenshots (highlighted as some examples) to show the difference between two revisions.

Surprisingly, even the number of editing sites and clusters changed dramatically, the "Avg.Cluster.Length", "Avg.ES.Cluster" and other columns are still same as those in the previous revision.

More surprisingly, the authors still claim that "There were ~4-5 times more RNA hyper-editing sites in MGE-GABA ($\mu=345,736$ sites) and GLU ($\mu=253,135$ sites) neurons than in OLIG ($\mu=66,077$ sites) (Cohen's $d = 2.64$, $p=2.6\times 10^{-6}$, linear regression)", "Overall, we identified 189,229 cell type-associated RNA editing sites, including a total of 107,998 sites on 4,781 genes in MGE- GABA..." in the results section.

The authors should be aware what they changed in Supplemental Table 1 are fundamental results (i.e. the number of RNA hyper-editing sites per sample), which will affect most (if not all) of the downstream results and conclusions. I can't believe the authors are not clear of this fundamental change.

Overall, I am disappointed that the authors revised these fundamental numbers without any explanations. Whatever mistakes they were, these are too critical to maintain the current results and conclusions valid. It also exposes a lack of rigor for the analysis procedure.

Reviewer #2 (Remarks to the Author):

I like to thank the authors taking efforts to address my remaining question. I have no more questions now and thus support the publication of this manuscript.

1) The authors have now provided the total number of reads and the alignment stats. However, I am surprised to see that the alignment ratio is always 100% for every sample, i.e. “PF_READS” equals to “PF_READS_ALIGNED” in Supplemental Table 1 – “RNAseq QC and alignment metrics” sheet, which is too good to be true.

Reply: PICARD tools was run on STAR mapped bam files without the unmapped reads, so perhaps unsurprisingly, the percentage of all pass filter reads is 100% (this was denoted in our supplemental table description file). However, as the reviewers main point of concern is nested in the hyper-editing analysis, which is applied to unmapped reads, we now include a set of PICARD tool statistics applied to bam files containing *all aligned and unaligned short reads*. Subsequently, while the total number of pass filter aligned reads and bases predictably remains unchanged for all samples, the total number of pass filter reads has increased and the *percentage of all pass filter reads aligned* is now ~87% across all samples with no significant variability by cell type.

2) I am very concerned that the authors revised some fundamental results/data in Supplemental Table 1 without any explanations. Furthermore, these revised results should lead to different downstream outcomes and/or conclusions. However, the authors only made changes in Supplemental Table 1, but not any other results in their study. For example, the number of total hyper editing sites (the “Total.hyperEditing.Sites” column) was revised from 331,780 to 31,779 for sample GABA-H395. That’s TEN times drop! In addition, this number was revised from 456,934 to 72,933 for sample GABA-H427. Please see the attachment for the screenshots (highlighted as some examples) to show the difference between two revisions.

Reply: Allow us to elaborate, and apologies for any confusion as we walk you through the concern. After the original request for further investigation, first author W.C. re-computed the hyper-editing analysis, and this left us unclear as to what specifically occurred in the original submission that uniquely altered these three values and not any of the other values for any other samples (e.g. coding issues, excel issues, copying issues, incomplete carry over of files *etc...*). We did previously amend **Figure 1D-E** and the corresponding statistics in accordance with this change. However, on the account of the highly diligent reviewer (which we are very grateful for), Dr. Breen has now repeated this analysis from scratch, starting with original raw fastq files, and has pinpointed two small, but relevant points of confusion, which are now fully rectified:

1. First, all hyper-editing sites were fully annotated by ANNOVAR, including whether a site mapped to a common SNP from dbSNP (maf > 0.05). However, such sites were improperly filtered, resulting in an increase of ~1000 hyper-editing sites/sample which were annotated as sites overlapping with common genetic variation. We have added the total number of hyper-editing sites filtered by a common SNP threshold per sample in **Table S1**. Although this has a tiny effect on the global hyper-editing rates per sample and by cell type, these sites should clearly be removed from the analysis and our interpretation – **this is fully resolved**.
2. With this aside, Dr. Breen reproduced the final hyper-editing counts per sample, but did identify disparities for samples H286-GABA and H427-GABA. *Upon deep inspection of the previous set of results*, we have identified an issue of recurring empty lines in the ANNOVAR output files for samples H286-GABA and H427-GABA, which previously impacted the total counts for these samples only and not for any of the other samples – **this is fully resolved**.

Surprisingly, even the number of editing sites and clusters changed dramatically, the “Avg.Cluster.Length”, “Avg.ES.Cluster” and other columns are still same as those in the previous revision.

Reply: We have now modified **Supplemental Table 1** to report an average cluster length of ~102bp, each containing ~14.4 hyper-editing sites on average (a decrease by ~0.1 sites per cluster as we previously reported) – **this is fully resolved**. Expectedly, these minor changes do not change the overall conclusions of our manuscript.

More surprisingly, the authors still claim that: “There were ~4-5 times more RNA hyper-editing sites in MGE-GABA ($\mu=345,736$ sites) and GLU ($\mu=253,135$ sites) neurons than in OLIG ($\mu=66,077$ sites) (Cohen’s $d = 2.64$, $p=2.6 \times 10^{-6}$, linear regression)”.

Reply: Yes, we thank the reviewer for bringing this to our attention. This should indeed be modified and now reads as: “*There were ~4 times more RNA hyper-editing sites in MGE-GABA ($\mu=266,621$ sites) and GLU ($\mu=251,790$ sites) neurons than in OLIG ($\mu=65,716$ sites) (Cohen’s $d = 1.83$, $p=0.0002$, linear regression) (Figure 1D). To minimize technical variability and facilitate a direct comparison across all cell types, we normalized the hyper-editing signal to the number of mapped bases per sample and again observed a preponderance of hyper-editing in neurons relative to OLIG (Cohen’s $d = 2.09$, $p=4.4 \times 10^{-5}$, linear regression) (Figure 1E).*” – **this is fully resolved**.

The authors still claim that: “Overall, we identified 189,229 cell type-associated RNA editing sites, including a total of 107,998 sites on 4,781 genes in MGE- GABA...” in the results section.

Reply: Actually, this sentence will not change. We do not include hyper-editing sites in any of the downstream analysis apart from **Figure 1 D-E** and **Supplemental Figure 1** because their supporting read coverage is often quite low. These numbers refer to selective RNA editing sites, as described/outlined in our results section headers (e.g. *Identification and annotation of bona fide site selective RNA editing sites*), result text, and materials and methods text.

The authors should be aware what they changed in Supplemental Table 1 are fundamental results (i.e. the number of RNA hyper-editing sites per sample), which will affect most (if not all) of the downstream results and conclusions. I can’t believe the authors are not clear of this fundamental change. Overall, I am disappointed that the authors revised these fundamental numbers without any explanations. Whatever mistakes they were, these are too critical to maintain the current results and conclusions valid. It also exposes a lack of rigor for the analysis procedure.

Reply: While we agree that we could have been clearer in our original explanation for the source of this misalignment in our previous communication, we respectfully disagree with the reviewer’s conclusion that our hyper-editing results impact downstream results and conclusions. It is important that the reviewer understands that it is completely inconceivable for the hyper-editing analysis to impact any part of our downstream analysis. Again, this is because we do not include hyper-edited reads/sites anywhere in this report apart from **Figure 1D-E** and **Supplemental Figure 1**. For clarity, the main objective of our hyper-editing analysis was to compliment and fully inform the observation that global RNA editing rates are significantly higher in neurons relative

to oligodendrocytes, based on selective editing (from mapped bam) and hyper-editing (from unmapped fastq). After this observation, we do not use the hyper-editing sites/reads for any downstream analysis for the reason that the supporting read coverage is often lower than for selective sites called from mapped bam files. Again, the remainder of our analyses (following Figure 1) are based on site selective editing sites called from mapped bam files with sufficient supporting read coverage and detection rates across the majority of samples within a cell type – analyses both led and performed by Dr. Breen. Let us reassure the reviewer, that the data presented throughout this manuscript are rigorously executed and carefully studied, and the hyper-editing analysis does not carry over to our downstream analyses in any form.

While the aforementioned manuscript updates do not change the overall conclusions for this work, we sincerely thank the reviewer for their insightful comments, which has specifically led changes in hyper-editing quantification as reflected in revised **Figure 1E** (see below) – results, which still strongly support an elevated rate of hyper-editing in neurons compared to oligodendrocytes.

Original submission:

Figure 1E. Neurons vs. oligodendrocytes (Cohen's $d=2.24$, $p=1.1 \times 10^{-5}$)

Current resubmission:

Figure 1E. Neurons vs. oligodendrocytes (Cohen's $d=2.09$, $p=4.4 \times 10^{-5}$)